**Data Availability Statement:** All data files (de-identified patients metadata and fastq. files of

# Metagenomic sequencing of the skin microbiota of the scalp predicting the risk of surgical site infections following surgery of traumatic brain injury in sub-Saharan Africa

**Hervé Monka Lekuya**[1,2]*, **David Patrick Kateete**[3], **Geofrey Olweny**[3], **Edgar Kigozi**[3], **Larrey Kasereka Kamabu**[1], **Safari Paterne Mudekereza**[4], **Rose Nantambi**[1], **Ronald Mbiine**[1], **Fredrick Makumbi**[5], **Stephen Cose**[6], **Jelle Vandersteene**[2], **Edward Baert**[2], **Jean-Pierre Okito Kalala**[2], **Moses Galukande**[1]

1 Department of Surgery/Neurosurgery, CHS, Makerere University, Kampala, Uganda, 2 Department of Neurosurgery/Human Structure & Repair, Ghent University, Ghent, Belgium, 3 Department of Molecular Biology, CHS, Makerere University, Kampala, Uganda, 4 Faculté of Médecine, Neurosurgery Unit, Université Catholique de Bukavu, Bukavu, DRC, 5 School of Public Health, Makerere University, Kampala, Uganda, 6 Medical Research Council, London School of Hygiene & Tropical Medicine, Entebbe, Uganda

\* lekuya.monka@mak.ac.ug

## Abstract

### Background

Surgical site infections (SSI) are a significant concern following traumatic brain injury (TBI) surgery and often stem from the skin's microbiota near the surgical site, allowing bacteria to penetrate deeper layers and potentially causing severe infections in the cranial cavity. This study investigated the relationship between scalp skin microbiota composition and the risk of SSI after TBI surgery in sub-Saharan Africa (SSA).

### Methods

This was a prospective cohort study, enrolling patients scheduled for TBI surgery. Sterile skin swabs were taken from the surrounding normal skin of the head and stored for analysis at -80°Celcius. Patients were monitored postoperatively for up to three months to detect any occurrences of SSI. 16S rRNA sequencing was used to analyze the skin microbiota composition, identifying different taxonomic microorganisms at the genus level. The analysis compared two groups: those who developed SSI and those who did not.

### Results

A total of 57 patients were included, mostly male (89.5%) with a mean age of 26.5 years, predominantly from urban areas in Uganda and victims of assault. Graphical visualization and metagenomic metrics analysis revealed differences in composition, richness, and evenness of skin microbiota within samples (α) or within the community (β), and showed specific taxa (phylum and genera) associated with either the group of SSI or the No SSI.

metagenomics) are available from the Dryad database (accession number(s) through the following link: https://datadryad.org/stash/share/Rs4D-4iIKTXxL9h864irJfCAHkkss8q1SwyJ-nRd5xE. with the unique DOI: (DOI): doi:10.5061/dryad.47d7wm3p2.

**Funding:** Makerere Research Innovation Funds (Mak RiF) from the Government of Uganda, and Special Research Funds (BOF funding) of the Ghent University from the Flemish Government under the DESTINE Study.

**Competing interests:** The authors have declared that no competing interests exist.

**Abbreviations:** DMM, Dirichlet multinomial machine learning method; DSF, Depressed skull fracture; GCS, Glasgow coma scale; LDA, Linear discriminant analysis; MNRH, Mulago National Referral Hospital; OTU, Operational taxonomic unit; PCoA, Principal component analysis; SSA, sub-Saharan Africa; SSI, Surgical site infection; TBI, Traumatic brain injury.

## Conclusions

Metagenomic sequencing analysis uncovered several baseline findings and trends regarding the skin microbiome's relationship with SSI risk. There is an association between scalp microbiota composition (abundancy and diversity) and SSI occurrence following TBI surgery in SSA. We hypothesize under reserve that the scalp microbiota dysbiosis could potentially be an independent predictor of the occurrence of SSI; we advocate for further studies with larger cohorts.

## Introduction

One of the post-operative challenges of the surgical management of traumatic brain injury (TBI) is the occurrence of surgical site infections (SSI) in sub-Saharan Africa (SSA) [1,2]. This post-infectious complication is the commonest morbidity that leads to subsequent postoperative mortality among TBI patients up to 3 months after the initial surgery, with a prolonged hospitalization stay, increased healthcare costs, and impaired patient outcomes [3,4]. The bacterial origin of the infection during surgical procedures is very complex. They can be endogenous, exogenous (contamination), or both. Frequently, the infection originates from the surrounding residual bacteria of the skin where the surgical incision is made [5,6]. The skin microbiota is made essentially of symbiotic bacteria and fungi that act as a physical barrier, and their interactions with the human host promote skin homeostasis and immune response [7]. However, the skin microbiota of the scalp may become the potential source of infections if its composition is disrupted, and/or simply the physical barrier is breached, and bacteria are dragged into the deep layers of the skin, eventually going deeper up to the cranium cavity, causing life-threatening infectious complications if not treated quickly and energetically [8]. This suggests that there may be a potential link between the skin microbiome modification and SSI outcomes mainly where there is a skin breach. Even after meticulous skin disinfection before the surgical incision, the skin residual bacteria are not eradicated and may enter the surgical wound upon cutting, especially when the incision is large [9]. Indeed, the scalp has a complex microbial community in addition to the density of hair follicles on a small surface, in addition to its sebaceous glands that are deep to the layers not addressed by surgical disinfection. Like during non-neurosurgical procedures, the disequilibrium of the skin microbiota has been incriminated as the principal source of SSI [5]. The skin microbiota of the scalp contains potential contributors to the development of SSI, yet the role of its composition in predicting post-operative infection risk in neurosurgery remains poorly understood. Indeed, neurosurgeons rely on the known systemic skin phenotypic microbiota for the use of antiseptic for disinfection of the surgical site during surgery. However, with the recent advancements in the knowledge of the body's regional variability of the skin microbiota, and the genotypic diversity of this flora, there is a need to identify the microbiota of the skin surrounding the head for the SSA population. This can identify the potential role of their skin microbiota in the incidence of SSI. Most of the previous studies were based on the commonest phenotypic identification from culture and sensitivity of the wound following SSI. Indeed, there is an equilibrium of those residual micro-organisms that prevent the colonization and development of an infectious process from the external bacteria if introduced in the layers of the skin. A 16S rRNA sequencing of the skin surrounding the head could be the best way to identify those organisms and re-orient the management of patients undergoing neurosurgical procedures from the

traumatic head injury, as well as other indications. Recently, there has been a body mapping of the skin microbiota where the scalp showed already a different composition in skin microbiota from the rest of the body [10]; each body region and cavity has a specific skin microbiota, and each person has relatively a different profile of the skin microbiota as is the case for the fingerprint. This study aimed to elucidate the relationship between the composition of the scalp skin microbiota and the risk of SSI after TBI surgery.

## Materials and methods

### Study design and setting

The study design was a prospective cohort study conducted at the Mulago National Referral Hospital (MNRH), Kampala, Uganda between 18 March 2021 to 28 February 2022. This research is a subset of the DESTINE-Study (protocol registered in the Uganda National Council of Science and Technology as HS1284ES) that focused on postoperative infectious outcomes of TBI patients with depressed skull fracture (DSF).

### Study participants

**Inclusion criteria.**   This present study involved a population of TBI patients of all ages exclusively with the diagnosis of DSF as documented on their admission brain CT scan at the admission of the Accident and Emergency Unit of MNRH or at the referring hospital within 6 hours of injury, with a post-resuscitation GCS above 8, with SpO2 > 94% in room air, hemodynamically stable, and whose informed written consent was obtained by themselves or by their legal next-of-kin.

**Exclusion criteria.**   We excluded patients with evidence of scalp infection, gross wound contamination, skin loss, or other signs of infections before surgery, patients re-admitted after an attempt of non-surgical management, and with a history of brain surgery, steroids treatment, or with comorbidities. We also excluded patients whose skin swab samples had failed the quality check before the 16 rRNA metagenomics sequencing.

**Study variables.**   The independent variables were: Skin microbiota composition (taxonomy and skin microbiome abundancy of the scalp), as well as the demographics. The dependent variables were the occurrence of SSI as defined by the CDC [11], and the microbiological findings in terms of culture-sensitivity patterns from wound isolates.

### Particapants sampling

This was by convenience from a fixed cohort within the DESTINE Study.

### Study procedure I: Clinical management and bacteriological studies

**Patients' recruitment and management.**   We enrolled patients scheduled for TBI surgery. They received routine trauma care from resuscitation up to the timing of surgery (analgesics, antibiotics, anti-epileptic drugs, and fluids). They all received peri-operative intravenous antibiotics prophylaxis during anesthesia induction; the dosage of the drugs with weight-adjusted dose for pediatric patients was given as follows: cefazoline 2g with a repeat in 3–4 hours of surgery, ceftriaxone 2g with a repeat in 3–4 hours of surgery, or occasionally vancomycin 15mg/kg, ampicillin-sulbactam 2g-1g in continuation or substitution of the pre-operative antibiotics treatment. They had surgery of DSF at different times from the injury time based on the referral status and the team readiness. Postoperatively, they also received additional intravenous antibiotic treatment in continuation or in adjustment in case of evidence of infection based on the results of the antibiotic susceptibility for the entire duration of the hospital stay. Patients

were followed up on the neurosurgical wards in routine care, then shifted to oral antibiotics, analgesics, and other antiepileptic drugs at discharge; they were then later reviewed in outpatient clinic every month for up to 3 months to record the occurrence of SSI.

**Clinical diagnosis of surgical site infection.**   During their hospital stay and in the outpatient clinic review, any occurrence of SSI infection was recorded. Wounds' clinical inspection was done during a change of dressing by the attending clinician and the research assistant using the cranial SSI criteria of CDC to diagnose the SSI [11]; evidence of infection for the following 3 months of surgery was recorded and a swab of any wound discharge or wound dehiscence was taken for microbiological analysis of culture and sensitivity at the microbiology laboratory of Makerere University Uganda, as well as a complete blood count to support the clinical suspicion. A follow-up brain CT scan was obtained if indicated to detect eventual intracranial infections.

**Laboratory investigations for microbiological culture identification assays and drug susceptibility testing following the clinical diagnosis of surgical site infection.**   Isolation and identification of microorganisms was done by the inoculation of the sample on plated chocolate blood agar and blood agar for Gram-positive bacteria and MacConkey agar for Gram-negative bacteria. The plates were then incubated in a 5–10% $CO_2$ incubator at 35–37 C degrees for 24–48 hours. Colonies were identified morphologically by the microbiologist using appropriate Gram staining. The standard disc diffusion technique for antimicrobial susceptibility testing was performed on Mueller Hinton agar using the guidelines of the Clinical and Laboratory Standard Institute (CLSI) [12]. Gram-positive microorganisms were tested using Cefoxitin, Chloramphenicol, Clindamycin, Erythromycin, Gentamicin, Oxacillin, Trimethoprim-Sulfamethoxazole, Tetracycline, and Vancomycin. Standard antimicrobial disks were set and incubated overnight at 37˚C. Gram-negative microorganisms were tested using Amikacin, Doxycycline, Gentamycin, Ceftazidime, Cefuroxime, Piperacillin/Tazobactam, and Meropenem. As for disk diffusion methods recommendations from the CLSI [13], and also tallying with the abacus used in the microbiology laboratory of Makerere University Uganda, each antibiotic tested with a specific dose (in µg/disk) has its inhibitory zone diameter in millimeter, classifying results of the antibiotics susceptibility of the disk as sensitive (above the upper cut-off value), intermediate (in between the 2 cut-off values), and resistant (the lowest cut-off value).

## Study procedure II: 16S rRNA metagenomics sequencing

**Collection of the skin swab for metagenomics.**   After obtaining the informed consent, during the perioperative period between study recruitment and anesthesia induction before surgery, a skin swab of the surrounding normal skin (e.g: retro-auricular skin at the hairline) was collected before the time of skin preparation and surgical prep (Fig 1). The retro-auricular hairline region was chosen due to the fact that is relatively less in contact with the hospital bed linens while the patient is lying supine on the bed for several hours. A sterile skin swab on the surrounding skin was taken using a sterile cotton swab after scrubbing that skin with normal saline solution, about 1 to 2 cm in diameter, and was thoroughly swabbed for 30–45 seconds to ensure adequate microbial collection, then taken to the Molecular Biology laboratory of Makerere University for microbiota analysis of the superficial skin layer.

**DNA extraction.**   This was carried out following the manufacturer's recommendations for the Qiagen QIAamp DNA Mini Kit.

**PCR amplification and taxonomic analysis.**   Primer design of the V3 and V4 regions of the 16S rRNA gene were targeted for bacterial community analysis with suitable forward and reverse primers. A PCR setup was done of the PCR reaction mixture containing the extracted DNA, target-16s V3V416S, amplicon PCR forward

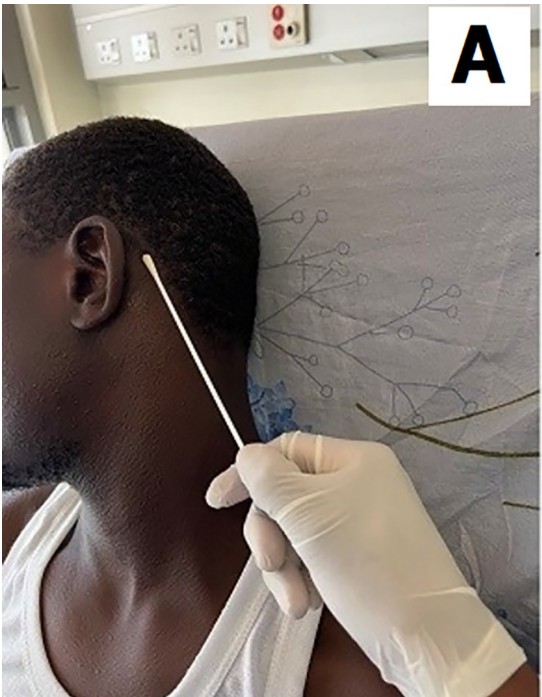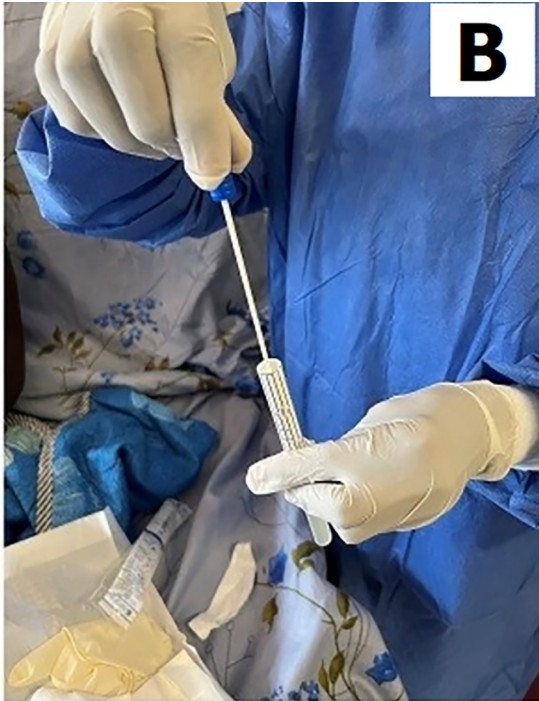

**Fig 1.** Illustration of normal skin swabbing: (A) Swabbing site at the scalp hairline behind the ear of the non-injured side; (B): Sterile transportation and preservation of the skin swab sample.

primer = 5'TCGTCGGCAGCGTCAGATGTGTATAAGAGACAGCCTACGGGNGGCWGCAG3' and 16S amplicon PCR reverse

primer = 5'GTCTCGTGGGCTCGGAGATGTGTATAAGAGACAGGACTACHVGGGTATCTAAT CC3', PCR buffer, sterile water, deoxynucleotide Tri Phosphates (dNTPs), and Taq DNA polymerase. PCR amplification was then performed in a thermal cycler, followed by the purification of PCR amplicons and then amplicon quantification.

**Library preparation.**   The size range of the amplicons was determined using gel electrophoresis and an amplicon size of 400–500 bp was selected for V3-V4 regions. End repair was performed on the purified amplicons to generate blunt-ended fragments suitable for adapter ligation through enzymatic treatment. Adenosine (A) nucleotide was added to the 3' ends of the repaired fragments using a polymerase enzyme and dATP to prepare the fragments for adapter ligation. Adapter ligation to the A-tailed fragments was then performed. A limited-cycle PCR amplification using primers that target the adapter sequences was performed to amplify the ligated fragments with attached adapters and barcodes. The amplified library was then purified to remove any remaining primers, adapters, and other PCR artifacts. Library quantification by qPCR followed to allow the pooling of equimolar amounts of different libraries for sequencing. Multiple indexed libraries (each with a unique barcode) were combined into a single pool, ensuring an equimolar representation of each library. Pooling multiple libraries allows for simultaneous sequencing and cost-effective use of the sequencing platform. The pooled library was submitted for sequencing on the Illumina MiSeq, sequencing machine (California, USA) model# M02903, serial number 410–1003. The library was loaded onto a flow cell, where clusters of DNA fragments were generated through bridge amplification. The sequencing-by-synthesis method generated raw sequencing data in the form of short reads (encoded fastq files).

**Bioinformatics.** After sequencing, the resulting data was subjected to bioinformatics analysis. Quality control checks were performed on the raw sequencing data by using the Fast-QC version 0.12.0 (https://www.bioinformatics.babraham.ac.uk/projects/fastqc/). Read pre-processing was then done by trimming low-quality bases, removing ambiguous bases, and discarding reads that are too short or contain sequencing artifacts. This was done using Cutadapt version 4.6 (https://cutadapt.readthedocs.io/en/stable/). Pre-processed reads were clustered into Amplicon Sequence Variants (ASVs) based on their sequence similarity in using the DADA2 package version 1.30.0 (https://bioconductor.org/packages/release/bioc/html/dada2.html) in R studio version 2023.09.1 (https://posit.co/download/rstudio-desktop). Taxonomic labels were then assigned to these Amplicon Sequence Variants in the DADA2 package that uses BLAST (https://blast.ncbi.nlm.nih.gov/Blast.cgi) and the SILVA database (https://www.arb-silva.de/) for taxonomic assignment according to the Silva 138.1 prokaryotic SSU taxonomic training data formatted for DADA2 (https://zenodo.org/records/4587955).

## Statistical analysis

Demographics and clinical data were entered into an Excel spreadsheet, cleaned, and exported to R studio version 2023.09.1 (https://posit.co/download/rstudio-desktop) for analysis. Numerical data were summarized using mean and range, whereas categorical data were summarized as frequencies and percentages. Fisher's exact test was used to check the difference between independent categorical variables. Positive culture and sensitivity of results of patients' wound samples were also reported. For the metagenomics statistical analysis, R Studio version 2023.09.1 (https://posit.co/download/rstudio-desktop) with associated packages was used. Descriptive visualization was reported on the relative abundancy of the skin microbiome of patients who develop SSI with positive cultured microorganisms. Alpha diversity metrics, including Shannon, Observed, Chao1, Simpson, Inverted Simpson, and Fisher indices [14] were calculated to assess microbial diversity and richness between the group that developed SSI (SSI group) and the one that did not develop SSI (No SSI group). This was done using the phyloseq package version 1.48.0 (https://bioconductor.org/packages/release/bioc/html/phyloseq.html). The statistical significance of differences between the two groups of SSI and No SSI was determined using a paired Wilcoxon test that was adjusted for multiple testing using the Benjamini-Hochberg's method at False Discovery Rate (FDR) <0.01 [15].

Beta diversity measures were also calculated using the phyloseq package version 1.48.0 (https://bioconductor.org/packages/release/bioc/html/phyloseq.html) to analyze the microbial community structure followed by Permutational multivariate analysis of variance (PERMANOVA) tests for statistical significance of differences in microbial community composition between groups of SSI and No SSI [16]. Principal Component Analysis (PCoA) [17] based on Bray-Curtis dissimilarities [18] was used to visualize these differences. A Dirichlet multinomial distribution of the genus relative abundance was used to model the distribution of these multinomial parameters across samples based on probability [19]. For microbiome network analysis, Spearman's rank correlation analysis was also performed [20]. Microbial taxa that co-occurred were considered positively associated, while mutually exclusive OTUs were negatively associated. The mean co-occurrence score is the average strength of positive associations between OTUs, calculated by taking the average of the correlation coefficients that were computed. Differential abundance analysis was conducted using DESeq2 version 1.43.1 (https://bioconductor.org/packages/release/bioc/html/DESeq2.html) and STAMP (2.1.3) (https://beikolab.cs.dal.ca/software/STAMP) to identify taxa significantly associated with the risk of SSI. An LDA_Micro function in r (version 4.2.3) was used to conduct Linear Discriminant Analysis (LDA) in a microbiome context and identified differentially abundant features

(OTUs) stratified by SSI and No SSI groups. The identified features were ranked and filtered based on significance thresholds. We visualized the output using bar plots depicting differential features stratified by SSI and No SSI at the genus level with LDA scores.

**Ethical consideration.** This study is a subset of the research project on the surgical timing of TBI patients (DESTINE-study) that obtained ethical approvals at all levels (S1 File) from the Makerere University School of Medicine Research and Ethical Committee(Mak SOMREC) as SM-2020-7, from the MNRH as an administrative hospital clearance, and from the Ugandan National Council of Science and Technology (UNCST) as HS1284ES. A written consent form (English or Luganda) was required and obtained from patients or the next-of-kin before recruitment, and confidentiality was paramount. They also consented for the publication of all the research materials.

## Results

An initial total of 127 patients with DSF were pre-enrolled from the main study of SSI outcomes from DSF from the DESTINE Study group in Makerere University with MNRH, Kampala, Uganda from 18 March 2021 to 28 February 2022. Only 57 met the inclusion criteria of this current study as described in the patients' flow chart (Fig 2). Swabbing of their scalp skin was done at admission preoperatively within a mean of 2 (±1.44) days of injury, but their metagenomic sequencing was performed later as one batch. Indeed, in addition to the study inclusion criteria, we convened with those 57 patients because their samples were amplifiable with a successful quality check for sequencing.

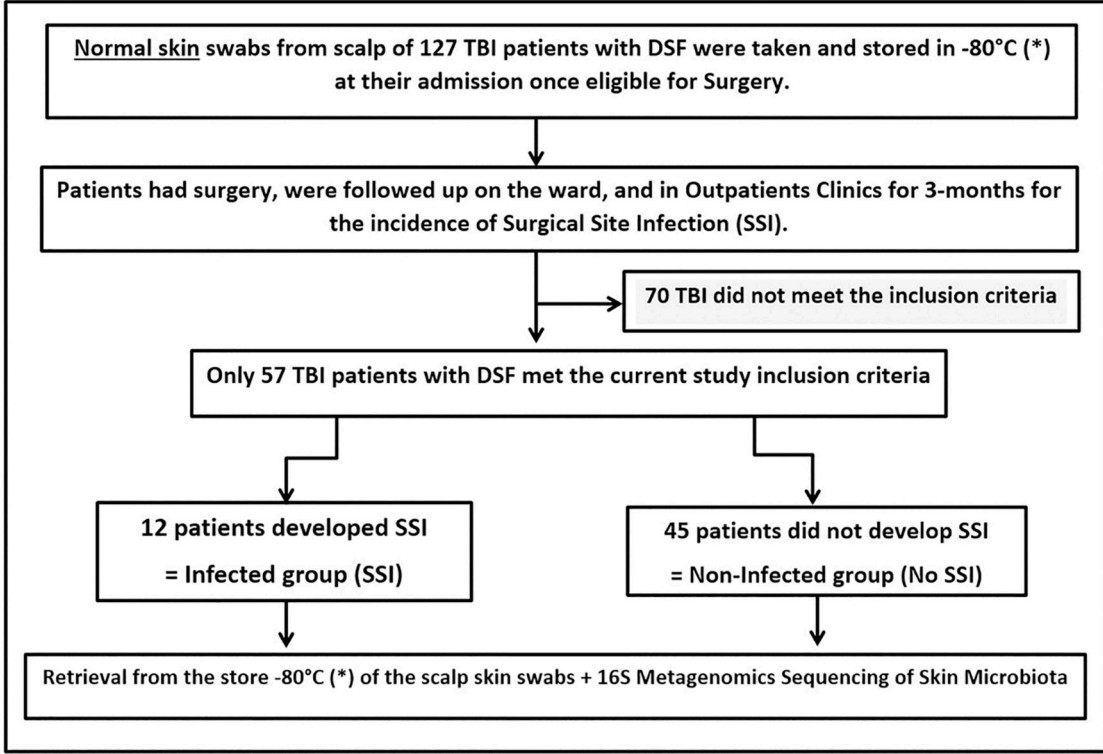

**Fig 2. Patients' flow chart.**

### Demographics, clinical, and bacteriological patterns of the patients

For the 57 patients, the mean age was 26.5 years and the majority of 89.5% were males. Most of them came from the urban areas of Uganda (56.1%), and were victims of assault (50.9%) as shown in Table 1.

There was an average of 51% simple and 49% compound DSF (Table 1).

The SSIs were mainly superficial incisional infections in 83.0%. The samples of the 12 patients with clinically diagnosed SSI underwent bacteriological studies of culture/sensibility and antibiotic susceptibility. We found that mono-bacterial infection of Gram-positive micro-organisms (*Staphylococcus aureus 2* + *Enterococcus spp* 4) constituted the highest number of isolates with 6 isolates among 12 patients with SSI (Table 2). Gram-positive isolates showed resistance to most of the commonly prescribed antibiotics. One case of mortality due to intra-cranial infection was attributed to poly-microorganism infections associated with *Escherichia coli* and *Klebsiella pneumoniae*. Two pus swab samples had no growth after 72 hours of micro-biological culture and antibiotic sensitivity.

### Analysis of the metagenomics sequencing

For the skin microbiome analysis of the sequencing, data revealed a diverse array of microbial species inhabiting the scalp microbiota in relative abundance in terms of age group at the level of phylum (Fig 3). The phyla of Gammaproteobacteria, Actinobacteria, and Bacilli vary in rela-tive abundance with the age groups from the pediatric sub-groups respectively from the below 5 years, then ≥ 5–11 years, and finally ≥12–17 years to an equilibrium status of the same pro-portion at adult age sub-group of ≥ 18–39 years, and reverse to the age group of ≥ 40 years. Taxonomic classification of bacteria varies by sex in absolute abundances across the sampled

**Table 1. Distribution of patients' baseline demographics and clinical type of injury by outcomes of the Surgical Site Infections.**

| Variable | Overall (Row%) N = 57 | Surgical Site Infection | | |
|---|---|---|---|---|
| | | YES (col%)n = 12 (21%) | NO (col%)n = 45 (79%) | p-value |
| | | | | Mann-Whitney test p |
| **Interval between injury & skin swab collection:** Mean (±SD) in days | 2.00 (±1.44) | 2.08 (±1.00) | 1.96 (±1.55) | 0.7767 |
| **Age** in years: Mean (range) in year | 26.5 (2–61) | 27.1 (3–53) | 26.4 (2–61) | 0.4196 |
| **Sex** | | | | Fisher's exact p |
| Female | 6 (10.5%) | 2 (33.3%) | 4 (66.7%) | 0.5960 |
| Male | 51 (89.5%) | 10 (19.6%) | 41 (80.4%) | |
| **Type of residence** | | | | |
| Rural | 25 (43.9%) | 6 (24.0%) | 19 (76.0%) | 0.7472 |
| Urban | 32 (56.1%) | 6 (18.8%) | 26 (81.2%) | |
| **Mechanism of injury** | | | | |
| Assault | 29 (50.9%) | 5 (17.2%) | 24 (82.8%) | 0.4779 |
| Pedestrian knocked RTC | 14 (24.6%) | 5 (35.7%) | 9 (64.3%) | |
| passenger motorcycle RTC | 8 (14.0%) | 1 (12.5%) | 7 (87.5%) | |
| Others | 6 (10.5%) | 1 (16.7%) | 5 (83.3%) | |
| **Clinical type of DSF** | | | | |
| Compound | 28 (49.1%) | 8 (28.6%) | 20 (71.4%) | 0.2070 |
| Simple | 29 (50.9%) | 4 (13.8%) | 25 (86.2%) | |

**Table 2. Distribution of microbial culture and antibiotic susceptibility among patients with SSI.**

| Patient code | Sex/Age | Isolated microorganism | Sensibility | Intermediate | Resistance |
|---|---|---|---|---|---|
| DSN 69242 | M, 38 yrs | Staphylococcus aureus | Vancomycin<br>Chloramphenicol<br>Linezolid<br>Rifampicin | - | Ciprofloxacin<br>Clindamycin<br>Erythromycin<br>Gentamicin<br>Penicillin G<br>Oxacillin<br>Amikacin |
| DSN 69257 | M, 21 yrs | Staphylococcus aureus | Linezolid<br>Rifampicin | - | Ciprofloxacin<br>Clindamycin<br>Erythromycin<br>Gentamicin<br>Penicillin Tetracycline<br>Oxacillin |
| DSN 69243 | M, 25 yrs | Enterococcus spp | Ampicillin<br>Chloramphenicol<br>High-L Gentamicin<br>Linezolid<br>Penicillin G<br>Rifampicin<br>Tetracycline | - | Erythromycin |
| DSN 69244 | M, 53 yrs | Enterococcus spp | Penicillin G | Erythromycin<br>Linezolid | Ampicillin<br>Chloramphenicol<br>Ciprofloxacin Vancomycin |
| DSN 69245 | F, 33 yrs | Enterococcus spp | High-L Gentamicin<br>Penicillin G | Erythromycin | - |
| DSN 69256 | M, 8 yrs | Enterococcus spp | High-level-Gentamicin | Ciprofloxacin Erythromycin Linezolid | PenicillinG<br>Vancomycin |
| DSN 69247 | M, 20 yrs | Acinetobacter spp | Amikacin<br>Imipenem | - | Cefepime<br>Gentamicin<br>Piperacillin<br>Tetracycline<br>Trimethoprim-Sulfamethoxazole |
| DSN 69248 | F, 22 yrs | Pseudomonas Spp | Colistin | Ciprofloxacin | Amikacin<br>Cefepime<br>Gentamicin<br>Imipenem<br>Piperacillin |
| DSN 69251 | M, 21 yrs | Escherichia coli | Imipenem | Amikacin | Cefuroxime<br>Chloramphenicol<br>Ceftazidime<br>Ciprofloxacin<br>Gentamicin<br>Trimethoprim-Sulfamethoxazole |
| DSN 69241 | M, 37 yrs (†) | Escherichia coli | Ciprofloxacin<br>Gentamicin<br>Imipenem | Peperacillintazobactam | Trimethoprim-SulfamethoxazoleAmpicillin<br>Ceftazidime<br>Ceftriaxone<br>Cefuroxime |
| | | Klebsiella pneumoniae | Ciprofloxacin<br>Gentamicin<br>Imipenem | - | Ampicillin<br>Ceftazidime<br>Ceftriaxone<br>Cefuroxime<br>Trimethoprim-Sulfamethoxazole<br>Peperacillin-Tazobactam |
| DSN 69252 | F, 36 yrs | Pus sample with no growth | Undetermined | Undetermined | Undetermined |
| DSN 69255 | M, 2 yrs | Pus sample with no growth | Undetermined | Undetermined | Undetermined |

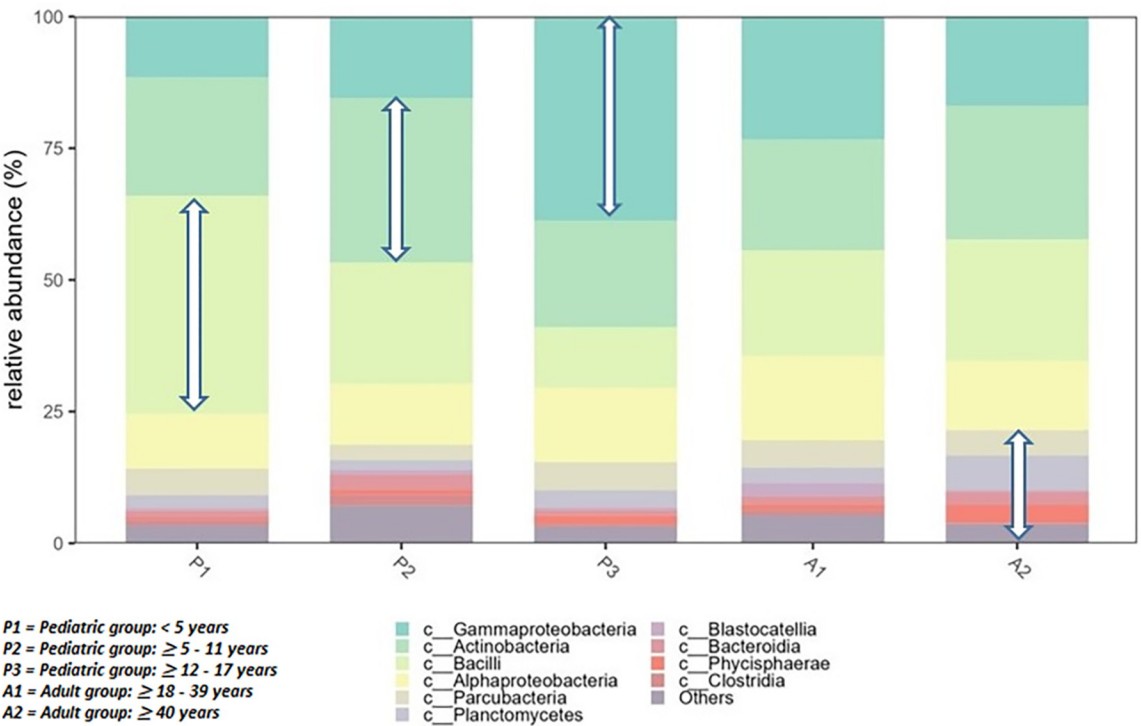

**Fig 3. Skin microbiota Phylum relative abundancy by age group.**

individuals in terms of absolute abundancy at the level of genus (Fig 4). There is a reverse abundancy relationship with the genus Corynebacterium versus Staphylococcus when comparing males and females in graphical visualization.

### Comparative analysis of the skin microbiota composition and the risk of SSI

When doing a comparative analysis of the scalp microbiota composition between the two groups of SSI and No SSI, there are differences in microbial diversity and absolute abundance in graphical visualization. Relative abundance analysis identified microbial genera that were associated with both the SSI and No SSI group (Fig 5). Patients in the SSI group exhibited a higher absolute abundance in the phylum of potentially pathogenic organisms, such as the Actinobateriota. Fig 6 shows the hierarchical clusters of individual samples of both SSI and No SSI and also Fig 7 shows a tree clustering in contrast to the visualization of the 4 major stacked bar plots of the phylum. Indeed, when merged in stacked bar plots, there is a reverse decreased proportion of proteobacteria and increased Actinobateriota among the SSI versus No SSI (Fig 8). In relative abundance, there is still a reversed proportion of microbial composition at the phylum level, as well as within the same phylum a reverse hierarchal abundancy of the genus of SSI versus No SSI. (Fig 9).

Fig 10 shows the comparisons of community alpha diversities between SSI and No SSI groups. The central line shown in each box plot indicates the median of the data (Wilcoxon rank-sum test). Furthermore, analysis of microbial diversity metrics of Shannon's α-diversity index of the microbiome shows a difference of 0.54 between the 2 groups of SSI and No SSI (Fig 11).

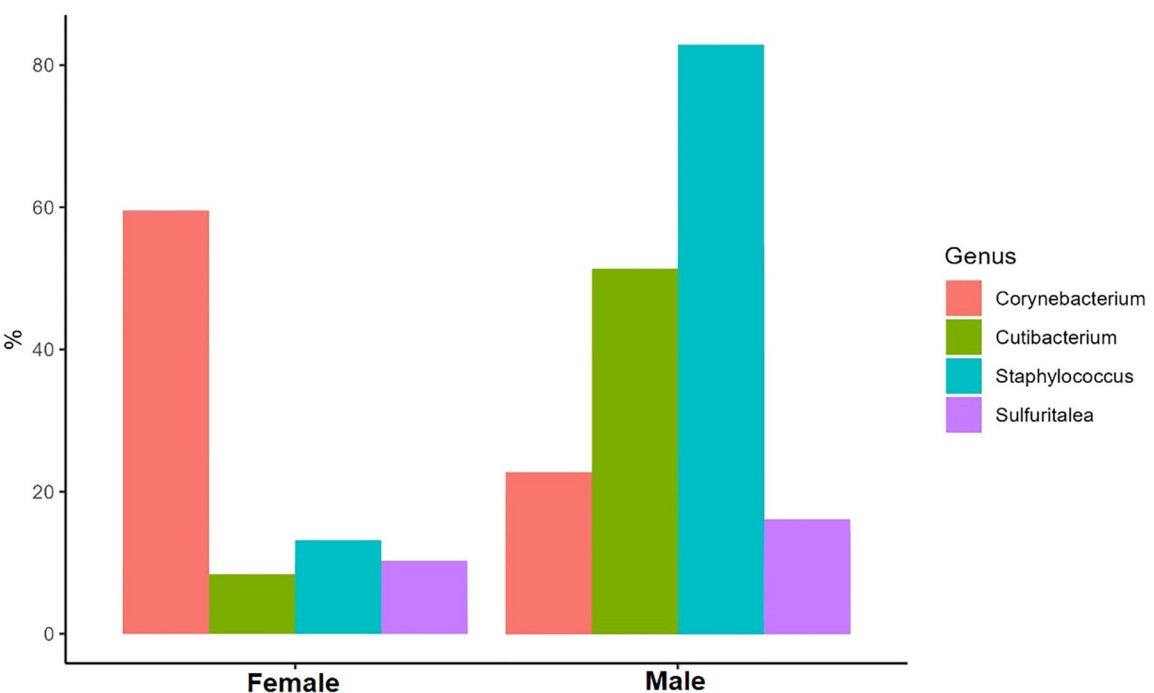

**Fig 4. Skin microbiota absolute abundancy by genera by sex.**

Beta diversity analysis showed a trend in clusters between microbial communities of patients with and without SSIs, indicating distinct compositional differences between the two groups (Fig 12). The PCoA plot with Bray-Curtis dissimilarity shows distances and clusters

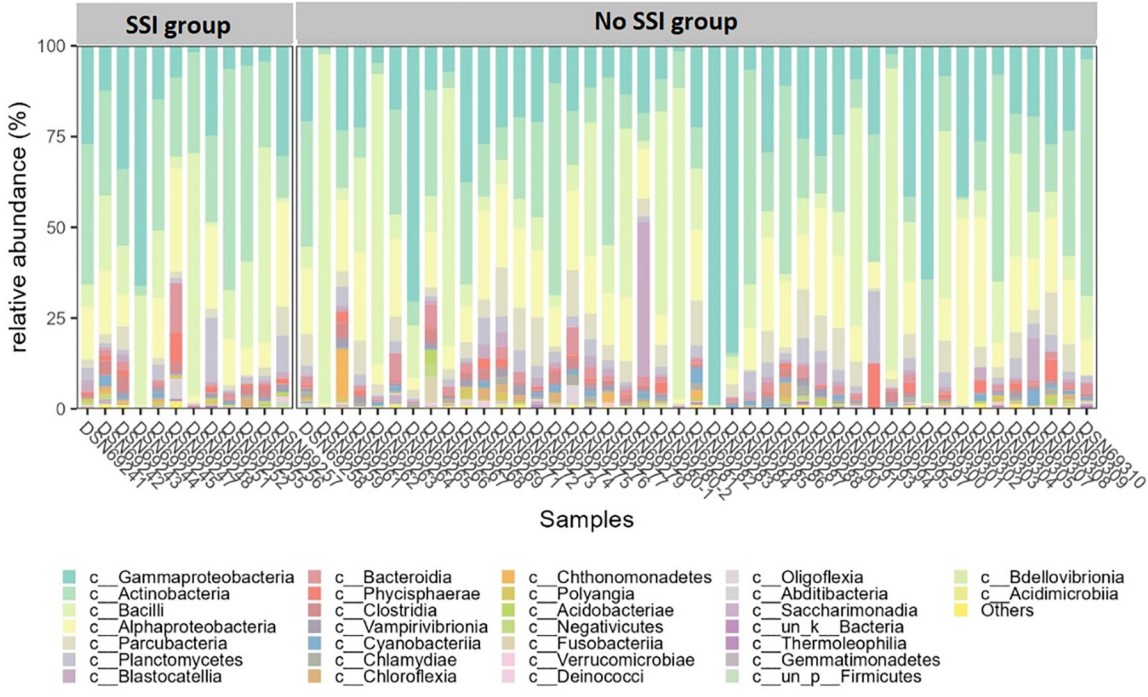

**Fig 5. Relative abundancy plot of both infected and non-infected groups by genera.**

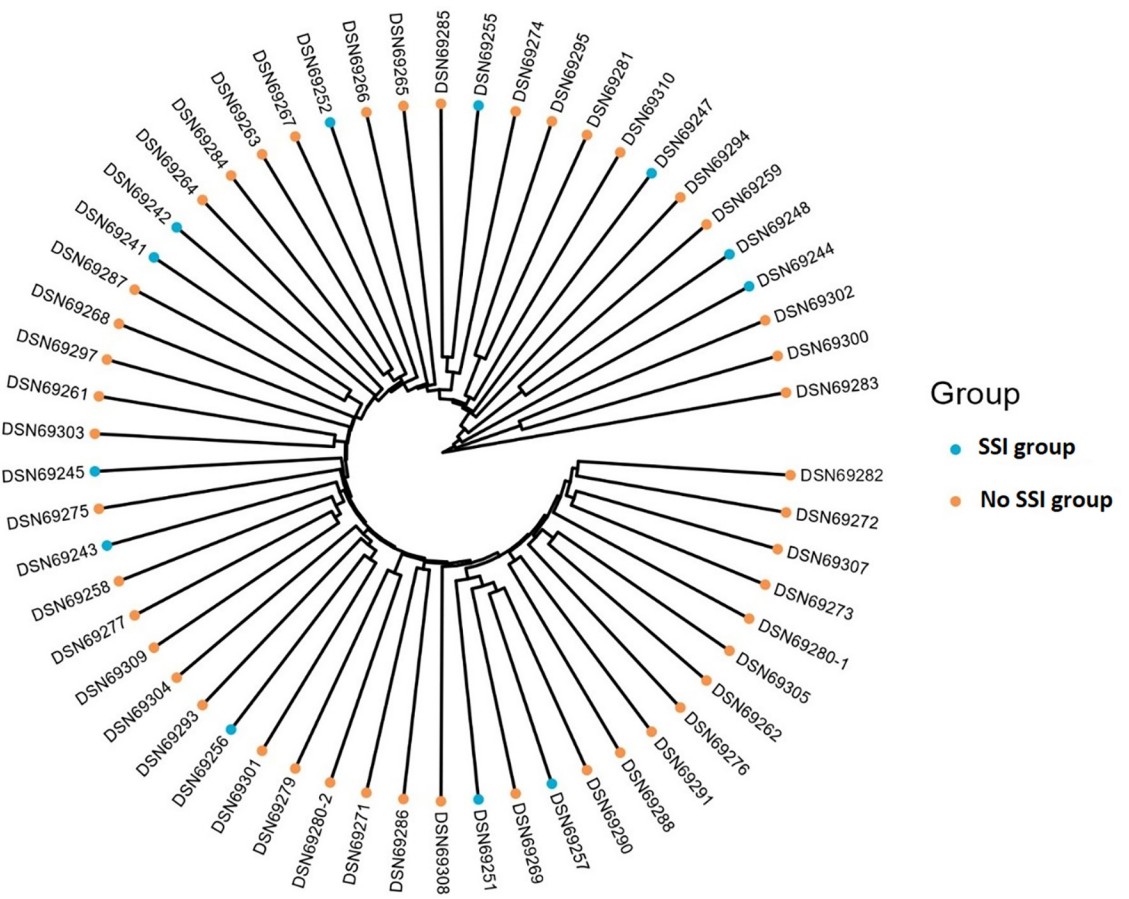

**Fig 6. Hierarchical cluster of infected and non-infected samples (Bray).**

between bacterial communities of individual samples from both groups, with PCoA1(13.22%), PCoA 2 (6.81%), Adonis: R 0.016 and a p = 0.785.

A Dirichlet multinomial machine learning model identified three microbial communities. The first community was predominantly composed of the Sulfuritalea and Cutibacterium genus. The second community was predominantly composed of the Staphylococcus and Sulfuritalea genura with the last community predominantly composed of the Acinetobacter genus (Fig 13). The beta diversity of these communities was not significantly different with the first community being a subset of the second community.

The STAMP differential genus analysis shows differences in relative abundance at the genus level between the SSI and No SSI (Fig 14). There were 30 differentiating genera in the SSI and No SSI groups, and clear differences were observed between the SSI and No SSI groups in terms of differential abundance up to the genus level. Stenotrophomonas, Sphingomonas, Enterococcus, Ochrobactrum, Massila, Novosphignobium, and Pseudomonas had a very low negative significant difference in mean populations of the No SSI group. Brachybacterium and Tepidisphera had a low positive difference in mean populations of the SSI group, thus, most associated with the occurrence of SSI in 95% CI.

Fig 15 shows a co-occurrence network for taxa in SSI versus No SSI groups at the phylum level; each node represents an OTU and blue lines show associations. An igraph analysis of occurrence was done with an alpha set at 0.05 for statistical significance. An igraph.degree

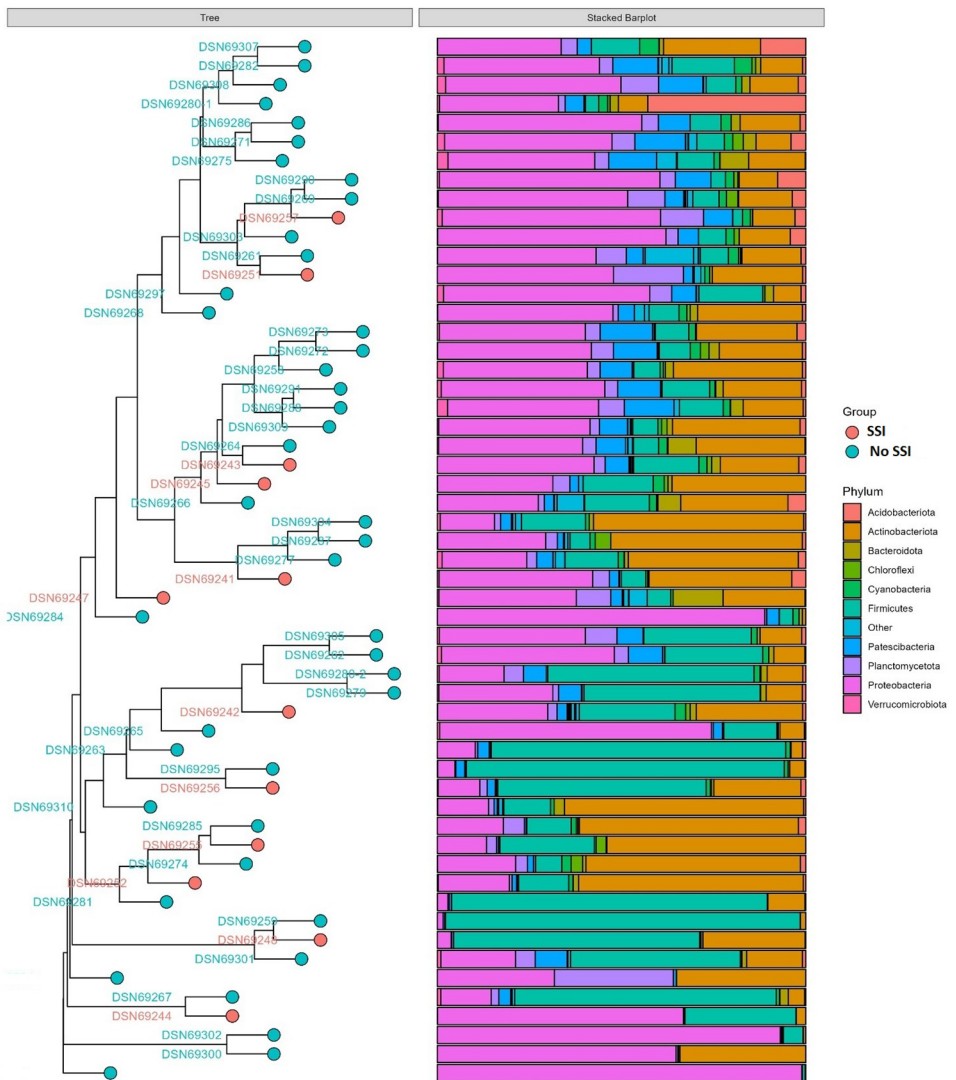

**Fig 7. Clustering tree of individual samples of both infected and non-infected patients by stacked bar plots of phylum.**

indicates the magnitude of correlations ranked at 20, 40, 60, and 80 respectively with an increase in diameter.

It was noted that the dense clusters of Proteobacteria, Firmicutes, and Actinobacteriota nodes were present in all networks (SSI and No SSI groups), but the network interactions were more noticed in the SSI group.

Bar plots show the linear discriminant analysis (LDA) effect size scores of OTUs analysis between SSI and No SSI groups (Fig 16); the LDA effect size (LEfSe) displays analysis between the two-group differences (SSI and No SSI) in skin microbial abundances. Specific effect sizes of significantly enriched taxa are highlighted on the cladogram and bar plot showing LDA scores stratified by the SSI group (green) and No SSI group (red). Phyla Verrucomicrobiota and Patescibacteria were indicated as biomarkers in the No SSI group. In the SSI group, Phyla Verrucomicrobiota and Proteobacteria were indicated as biomarkers.

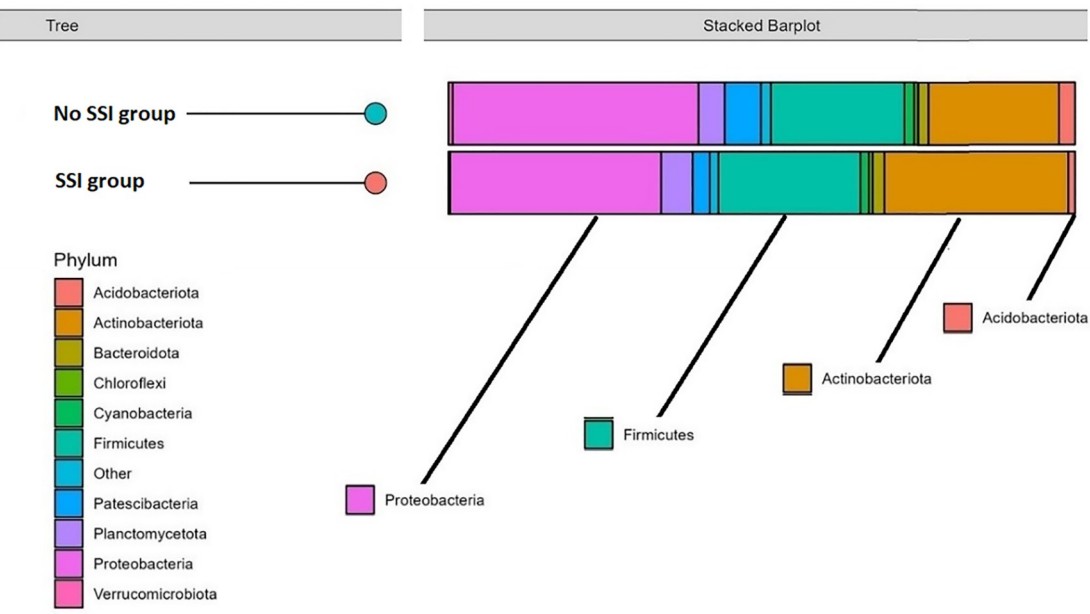

**Fig 8. Absolute abundancy bar plot of both infected and non-infected groups by genera.**

In the sub-analysis of skin microbiota of patients by isolated microorganisms with SSI, there is a disparity in microbial richness with the SSI of Acinetobacter (higher) and Pseudomonas (lower) in contrast to the rest (Fig 17). In general visualization, most of them have reduced abundancy in staphylococcus, with an increased abundancy of the 3 others (Fig 18).

## Discussions

This study was set up to elucidate the relationship between the composition of the scalp skin microbiota and the risk of SSI after TBI surgery. We navigated through the 57 patients recruited, with 12 who had SSI and 45 who did not have SSI after 3 months.

The participants were young males in the majority as is commonly seen in the trauma population group in SSA. In our study, most of the types of SSI were superficial incisional infections, and also mono-bacterial infections as seen also in the literature [21–23]. We found a higher antimicrobial resistance to common antibiotics, and it is well known from microbiological studies on neurotrauma patients in the hospital setting [24,25].

We observed an overall pattern of microbial species inhabiting the scalp, consistent with previous studies highlighting the complexity of the skin microbiota in a normal skin bacterial flora including *Staphylococcus*, *Corynebacterium*, *Propionibacterium*, *Streptococcus*, and *Pseudomonas* are part of the cutaneous microbiota [6,10]. Indeed, in our metagenomics sequencing study, we found a similar top 2 genera in terms of abundance of microorganisms *Staphylococcus*, *Corynebacterium* but in reverse order looking at the female sex. This may be due to the frequent use of hair treatment beauty products among young African females and may suppress and even promote other resistant forms. In addition, we found *Cultibacterium and Sulfuritalea* as the 3rd and 4th abundant genera respectively in the scalp skin. The scalp skin microbiota relative abundancy seems to change within the pediatric group and becomes almost the same after the age of 12 years as in the adult age. Our metagenomics sequencing analysis revealed several baseline findings and trends to support a relationship between the skin microbiome and the risk of SSI following TBI surgery, especially as a potential predictor nature.

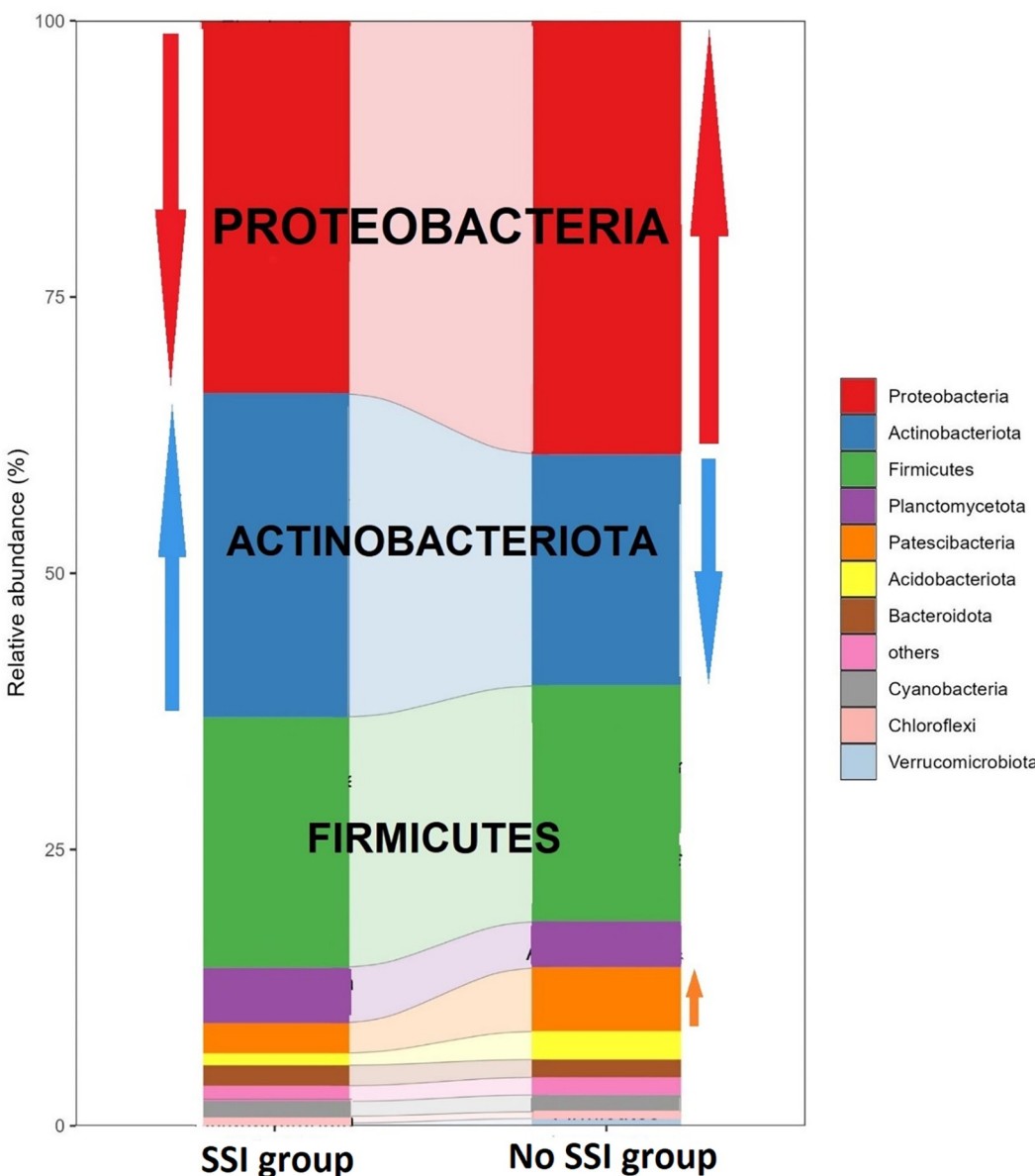

**Fig 9. Relative abundancy bar plot of both infected and non-infected groups by genera.**

When comparing both SSI and No SSI groups, there appears a significant difference in abundancy as well as in taxonomy. The Alpha diversity box plot of both SSI and No SSI groups reveals a difference when compared with the Shannon and observed within the 2 groups. This suggests a loss of microbial diversity and ecosystem stability in the scalp microbiota of individuals who developed the SSI. The Dirichlet multinomial machine learning method (DMM) (a generalization of the Multinomial distribution) is commonly used to model the distribution of counts for categorical data. In the case of microbial metagenomics, each sample (e.g., a DNA sequence read) can be thought of as a draw from a multinomial distribution, where each category corresponds to a different microbial species or operational taxonomic unit (OTU). The DMM model is a mixture model, which means it assumes that the observed data (e.g., sequencing reads) are generated from a mixture of multiple underlying components or clusters. In the

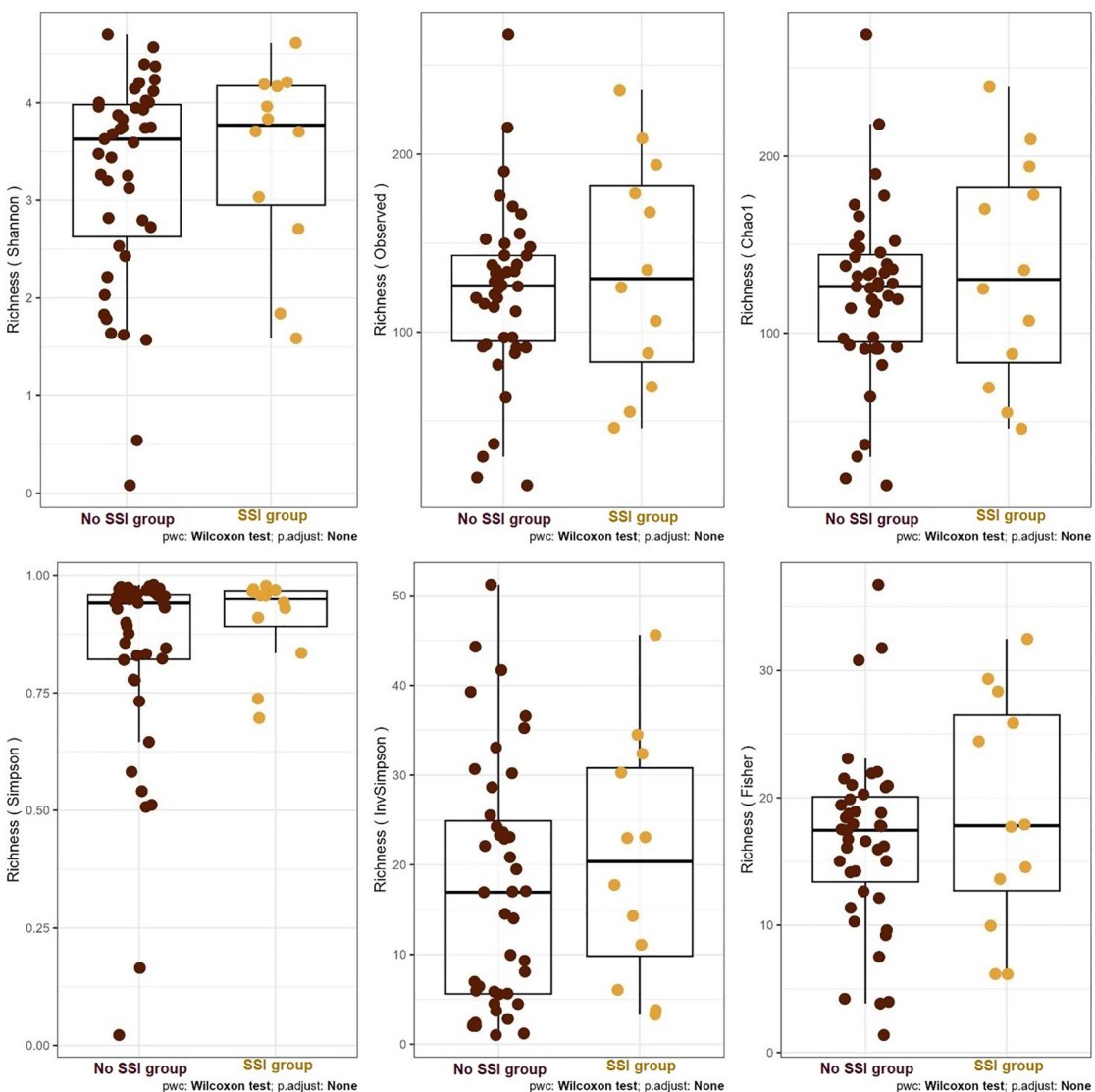

**Fig 10. Alpha diversity box plot of both infected and non-infected groups.**

context of microbial metagenomics, these components could represent different microbial species or community states. The mixture model framework allows for capturing the heterogeneity and complexity of microbial communities. The DMM model describes a generative process for how the observed sequencing data are produced [26]. By fitting the DMM model to observed sequencing data of SSI versus No SSI. We inferred the underlying composition of their microbial communities varies across different environments or conditions. When analyzing clusters by overall microbial community density in DMM, there were 3 major clusters composition with a higher abundancy of genera with Sulfiritalea, Staphylococcus, and Acinetobacter respectively. The equilibrium of complex human–microbe, and microbe-microbe interactions that exist on the surface of human skin illustrate the protective role of the microbiota, much like that of the gut microflora [6]. Patients who did not develop SSI showed a more balanced and stable microbiome like in the general skin microbiome as described by Egert *et al.* [10], characterized by a relatively higher abundancy in commensal bacteria and

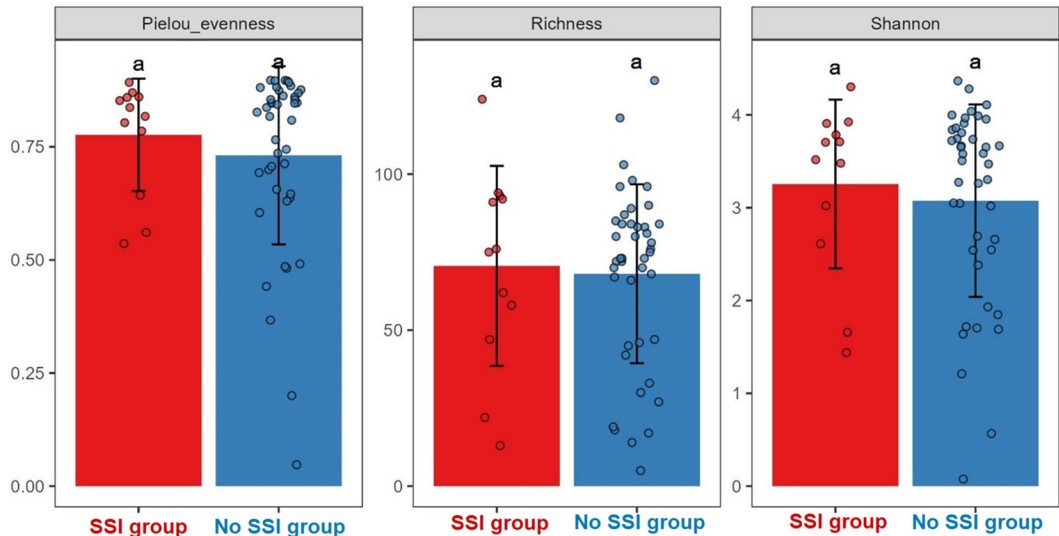

**Fig 11. Alpha-facet box bar of evenness, richness, and Shannon diversity.**

lower pathogenic ones in terms of their phylum. This highlights the potential balance of microbial diversity and ecological properties in maintaining skin health and preventing SSIs. This may explain why the physical interactions or modifications of the skin microbiota such as the local skin temperature, age, and environmental changes are the drivers of the SSI from skin breach. In our study, there were clear differences in the abundance of certain microbial genera between the SSI and No SSI groups. In breaking down the interpretation of the findings from the low negative significant difference in the No SSI group, we have the following genera: Stenotrophomonas, Sphingomonas, Enterococcus, Ochrobactrum, Massilia, Novosphingobium, and Pseudomonas, thus, protective to the occurrence of SSI. These genera have significantly lower mean populations in the No SSI group compared to the SSI group. A decrease in the abundance of these genera could indicate a disruption of the normal microbial community composition in individuals who developed the SSI. It is that there is a reverse way in which the predisposing infectious process has possibly altered the microbial environment, leading to a reduction in these genera, but this is very unlikely since the skin microbiome has been collected before the development of the SSI; so, the interpretation could vary. Regarding the low positive difference in the SSI group, conversely, Brachybacterium and Tepidiphilus showed a low positive difference in mean populations in the SSI group compared to the No SSI group. This suggests that these genera are more abundant in individuals who developed the SSI compared to those who did not. An increase in the abundance of these genera could be indicative of microbial dysbiosis associated with the infection. It is possible that a favorable environment promoted the proliferation of these genera or that they play a role in the pathogenesis of the infection. The microbial community composition differs significantly between SSI and No SSI individuals, particularly at the genus level. The identified genera may serve as potential biomarkers or indicators of infection, and further research is warranted to understand their roles in infection dynamics, host-pathogen interactions, and potential therapeutic interventions. Our study identified the skin microbiome of patients who developed SSI with multidrug-resistant microorganisms, providing insights into the mechanisms through which certain microbes may contribute to SSIs. Our findings were consistent with other studies, especially the significance of the genus Staphylococcus in SSIs, emphasizing its role as a common pathogen in the

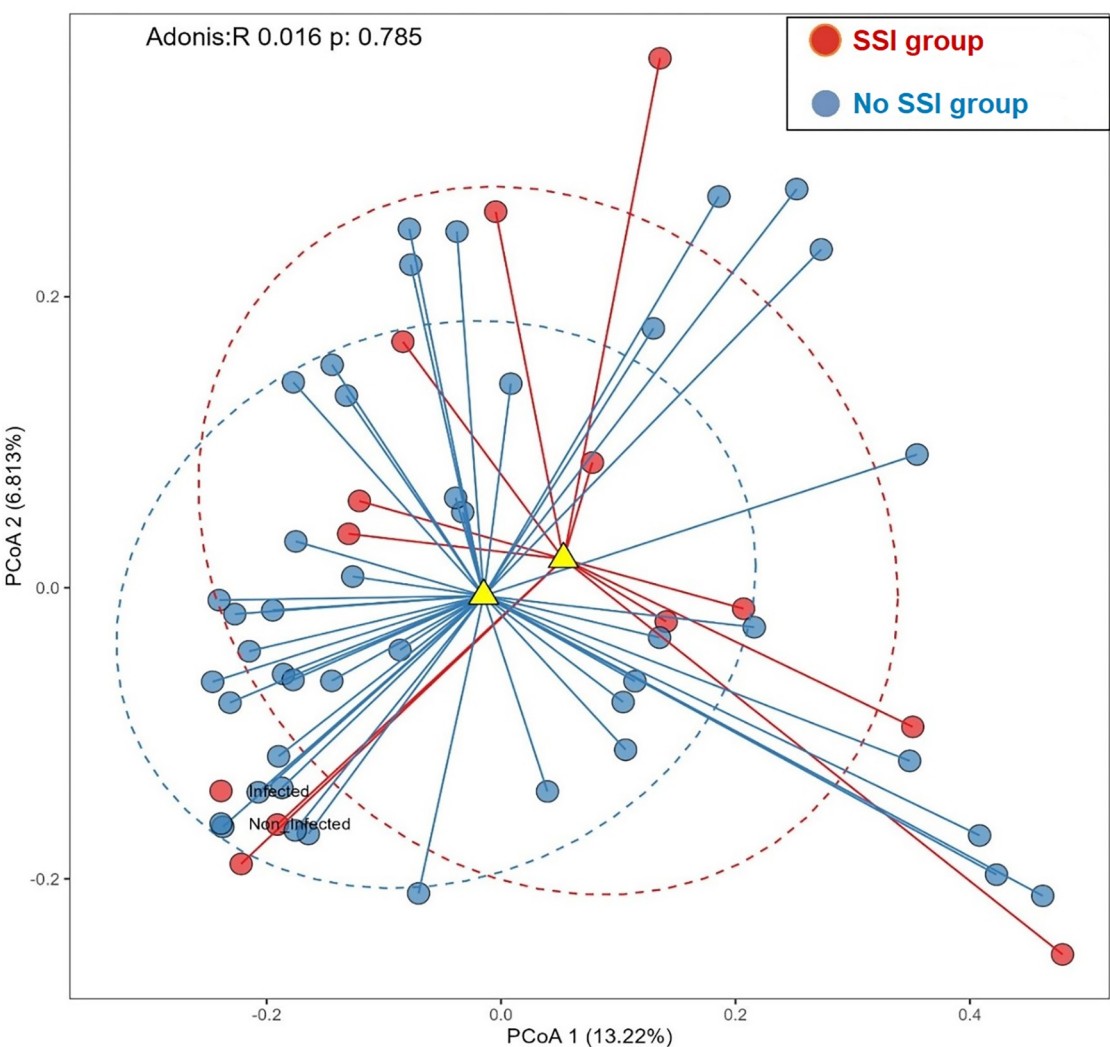

**Fig 12. Principal coordinate analysis (PCoA) of the plot with Bray-Curtis dissimilarity.**

infection process. TBI Patients who undergo emergency neurosurgical intervention in SSA may take several hours to days (referral to neurosurgery, brain CT scan, theatre space, etc.) without adequate incisional site preparation on the skin, and again this is worsened by a complex environment of the densely hairy region of the head. In addition, the head position on the hospital mattress linens, frequent bandaging, and additional pre-operative hair removal, skin contusion, or bruises can constitute additional factors. For example, it is still a debate whether hair removal with clippers before surgery reduces or not the risk of SSI infections, or whether the timing of hair removal influences the occurrence of SSI, and also it is known that the types of scalp differ from one race to another. The surgical practice relies mainly on antiseptic scrubbing solutions on surgical sites. Additionally, a consideration of the clinical contexts of SSI is essential for a comprehensive interpretation of these findings. It is a routine that most of those patients receive strong prophylaxis antibiotics, and this contributes to the effacement of the natural progress of the infection process when the skin breach is made in a contuse scalp. Thus, the patterns of the metagenomics sequencing of the scalp may be an independent factor in the incidence of SSI, in addition to the influences of extrinsic factors. Overall, most of our

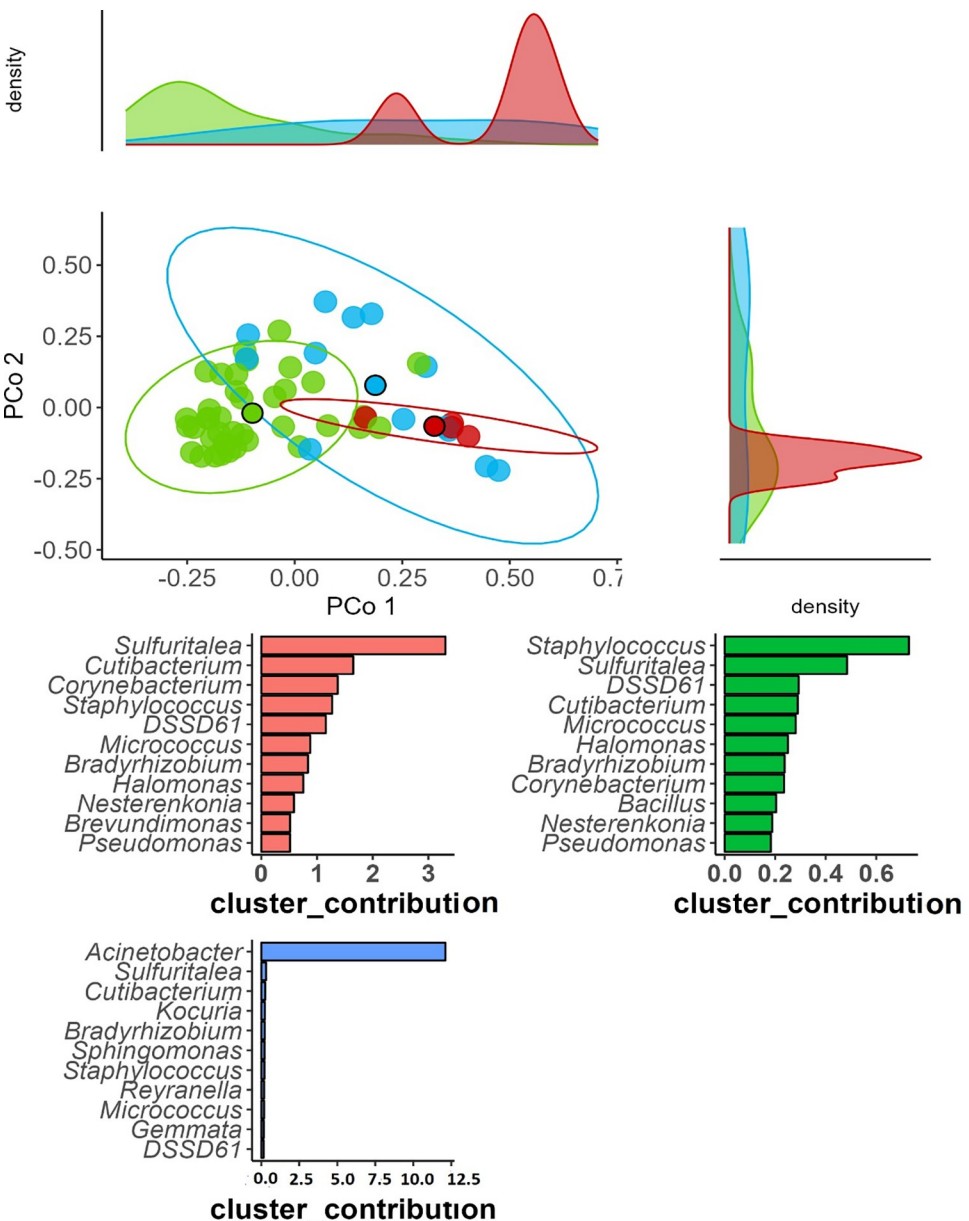

**Fig 13. Dirichlet multinomial machine learning by clusters of microbial community density.**

findings and visualization of the 2 groups suggest that the composition and diversity of the scalp microbiota can be predictive biomarkers for the risk of developing SSIs following cranial surgery for TBI; this highlights the potential role of scalp microbiota "dysbiosis" as the main underlying disruptive mechanism and predisposing patients to SSI as the scalp skin is a very complex with an extensive vascular network. The uniqueness of this research is that we collected both the incriminated micro-organisms from scalp SSI in patients and also reported the normal genotypic skin microbiota of the scalp from the SSA population. Thus, it gives information not only about the commonly found microorganisms of the skin of the scalp of the general population of SSA but also predicts which microbiota profile is more prone or protective against skin infections.

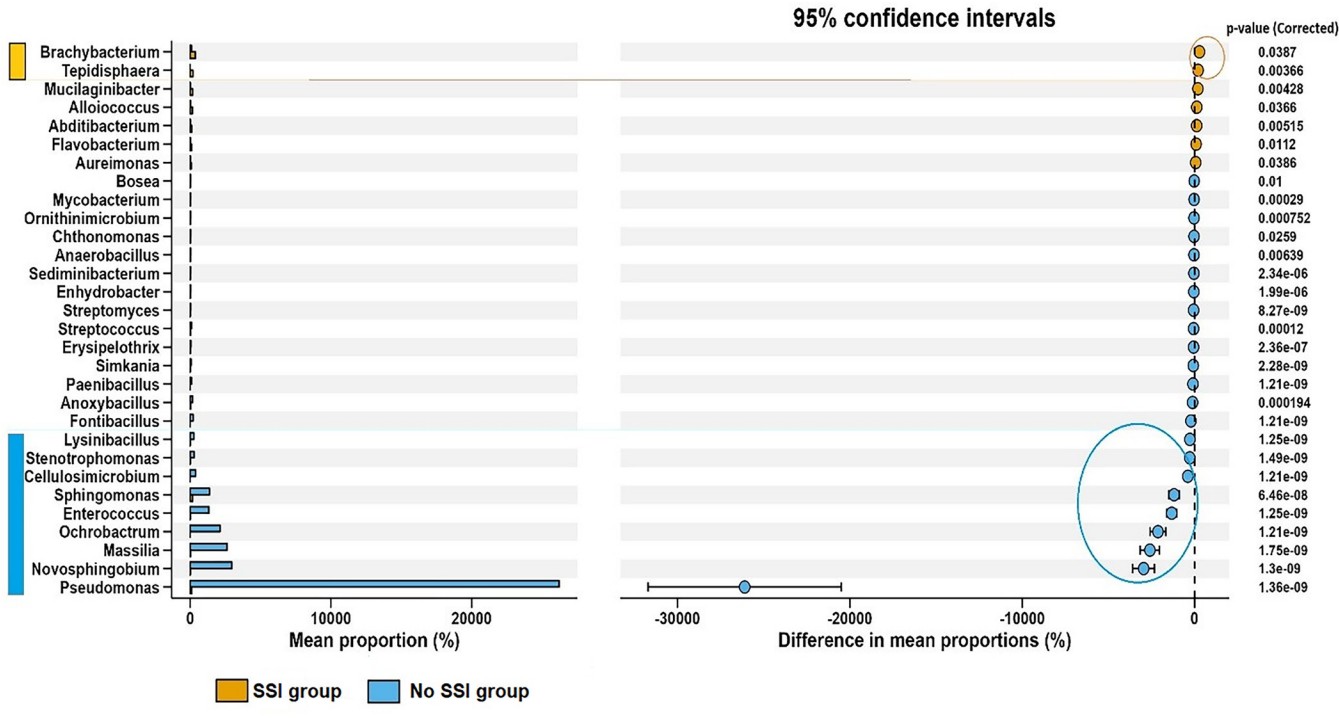

**Fig 14. STAMP differential analysis showing abundance at the genus level.**

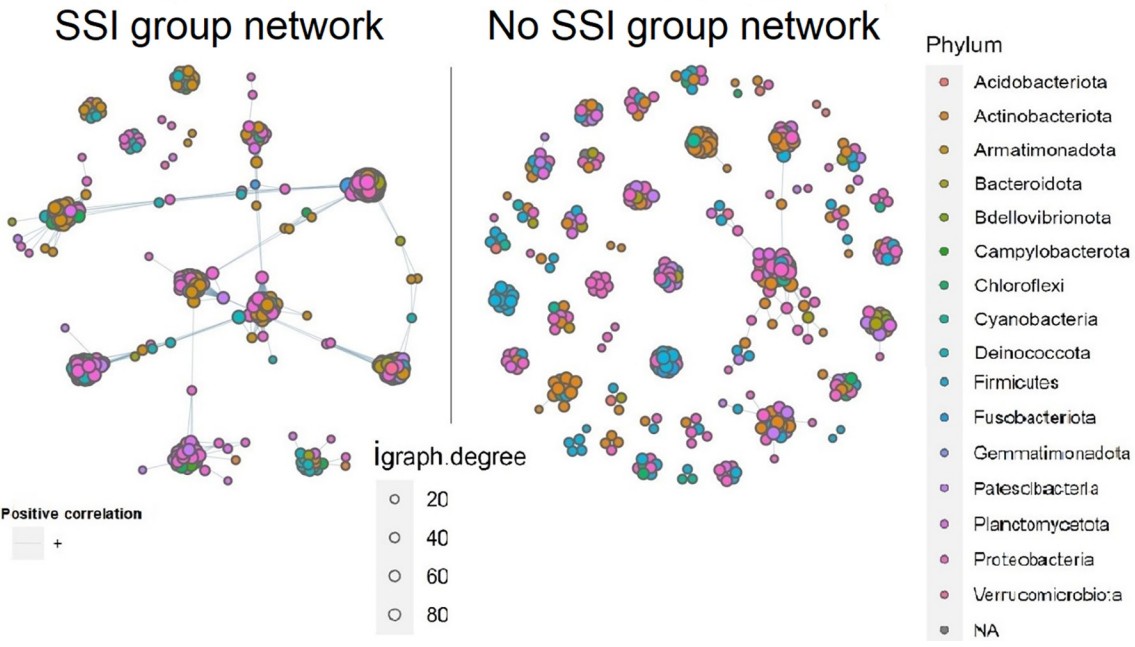

**Fig 15. Networking and proportion of nodes (OTUs color-colored) per phylum found in each cluster of microbial taxonomic composition between infected and non-infected groups.**

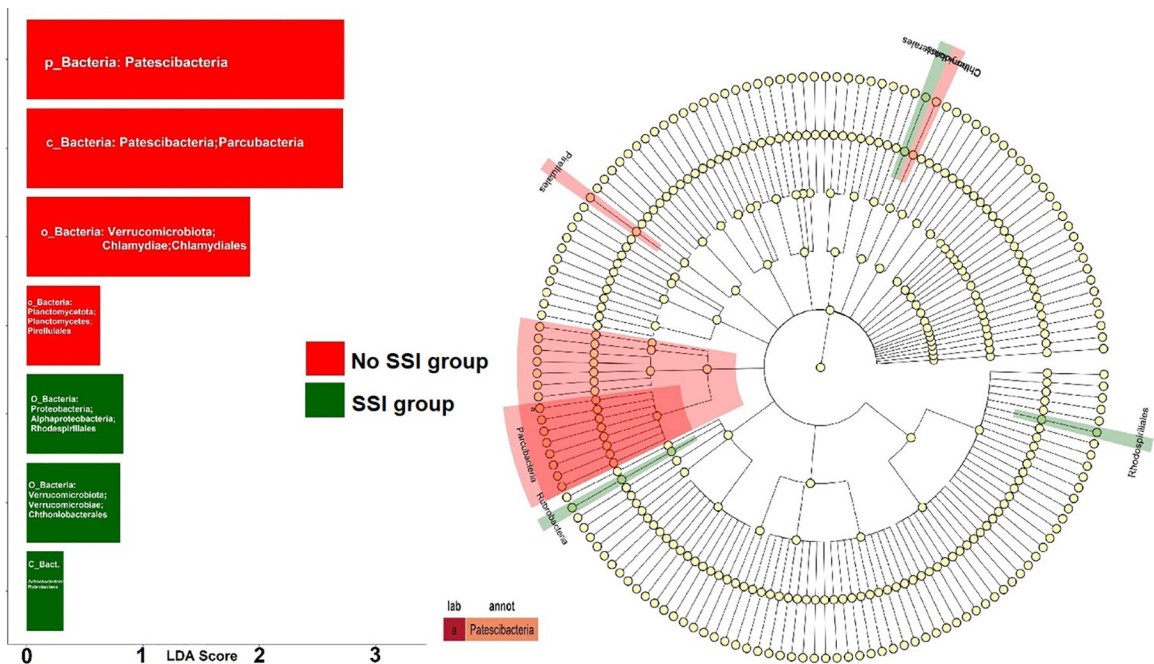

**Fig 16. LDA effect size (LEfSe) analysis of differences in skin microbial abundances between the two groups of infected and non-infected.**

Indeed, this study has relevance and potential clinical implications because it highlighted the composition and diversity of the skin microbiota on the scalp and its likelihood of predicting SSI following TBI surgery in SSA. The findings of this study can serve as a baseline of translational medicine by understanding the individual's skin microbiota profile, leading to tailored preventive strategies for SSI. It can also adjust the infection control practices in operating theatres by mapping the skin microbiota of high-risk patients and the targeted population (pediatric groups, etc.).

We acknowledge some limitations in our study; we only looked at the presence of bacteria in the skin microbiota and did not include fungi and viruses as part of the microbiota.

It is not excluded that our patients in the study might have had a degree of environmental modification of the skin microbiome composition due to several factors (transient contamination, skin moisture or temperature, etc.), especially after several hours or days following head injury. However, as mentioned in our methods, we attempted to swab the unaffected normal skin and rigorously reduced the superficial contaminations (sands, etc.) by cleaning only with normal saline to preserve also the inherent microbiota embedded in the superficial layers of the epidermis.

We had a relatively small sample size in a single-center and we did not account for potential confounders in the analysis. In our effort to minimize a multifactorial analysis with a smaller sample size, we included patients with better wound classification, higher GCS, no concurrent extra-cranial infections, and Oxygen saturation above 94% in room air as they are known factors to the development of SSI [27–29]. Despite its limitations, our study has postulated novel insights into understanding the relationship between the skin microbiome composition and the risk of SSI following TBI surgery in SSA.

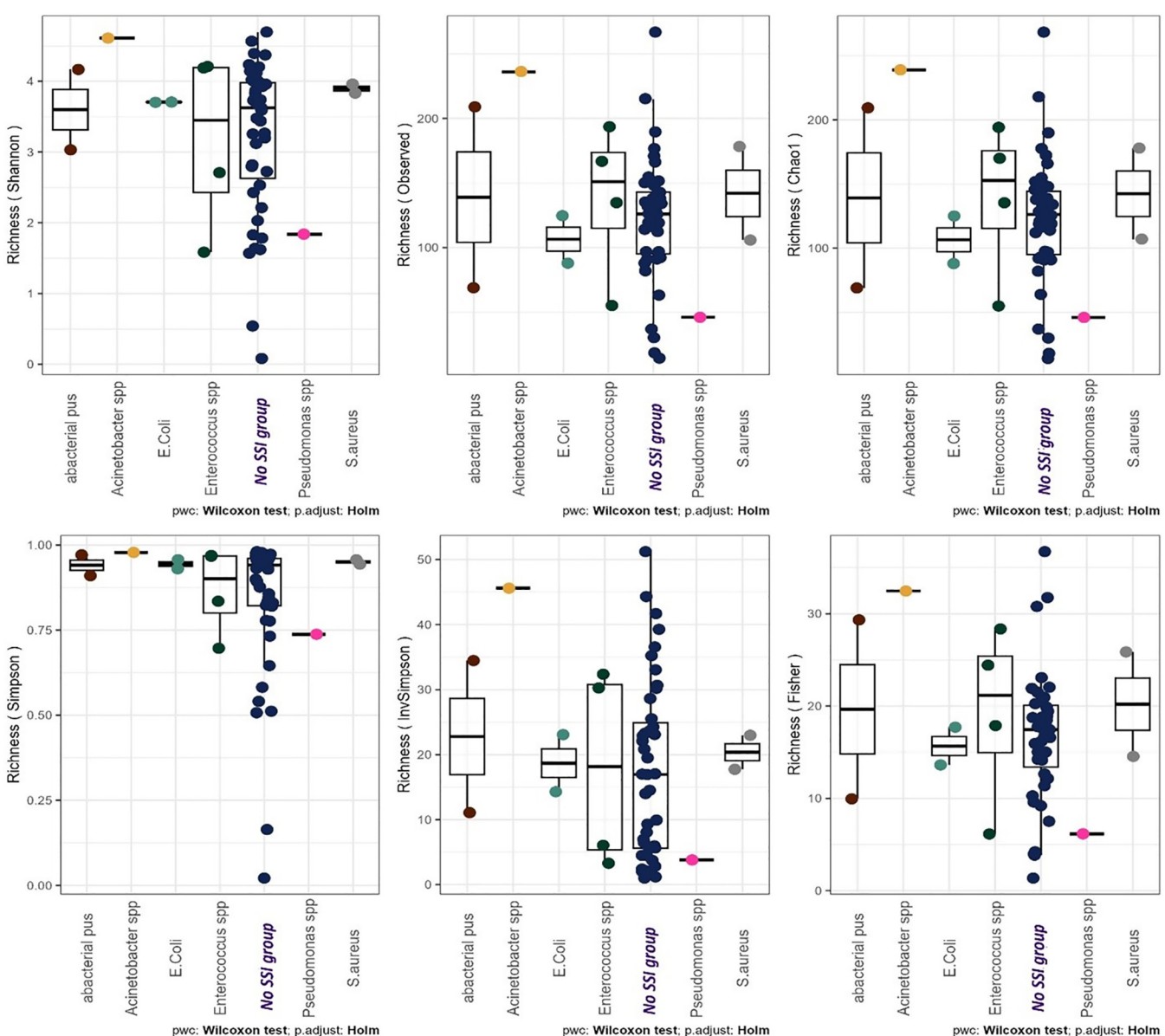

**Fig 17. Alpha cowplot diversity of patients' skin microbiota by isolated microorganism in the Surgical Site Infection.**

## Conclusion

The metagenomic sequencing analysis uncovered several baseline findings and trends regarding the skin microbiome's relationship with SSI risk. There is an association between scalp microbiota composition (abundancy and diversity) and SSI occurrence following TBI surgery in SSA. We hypothesize under reserve that scalp microbiota dysbiosis could be an independent predictor of the occurrence of SSI. This may vary with extrinsic factors such as skin temperature, pH, and environmental interactions. Further investigation and validation in larger multi-center cohorts is warranted to confirm the generalizability of these findings, but also to elucidate the underlying mechanisms driving this potential association.

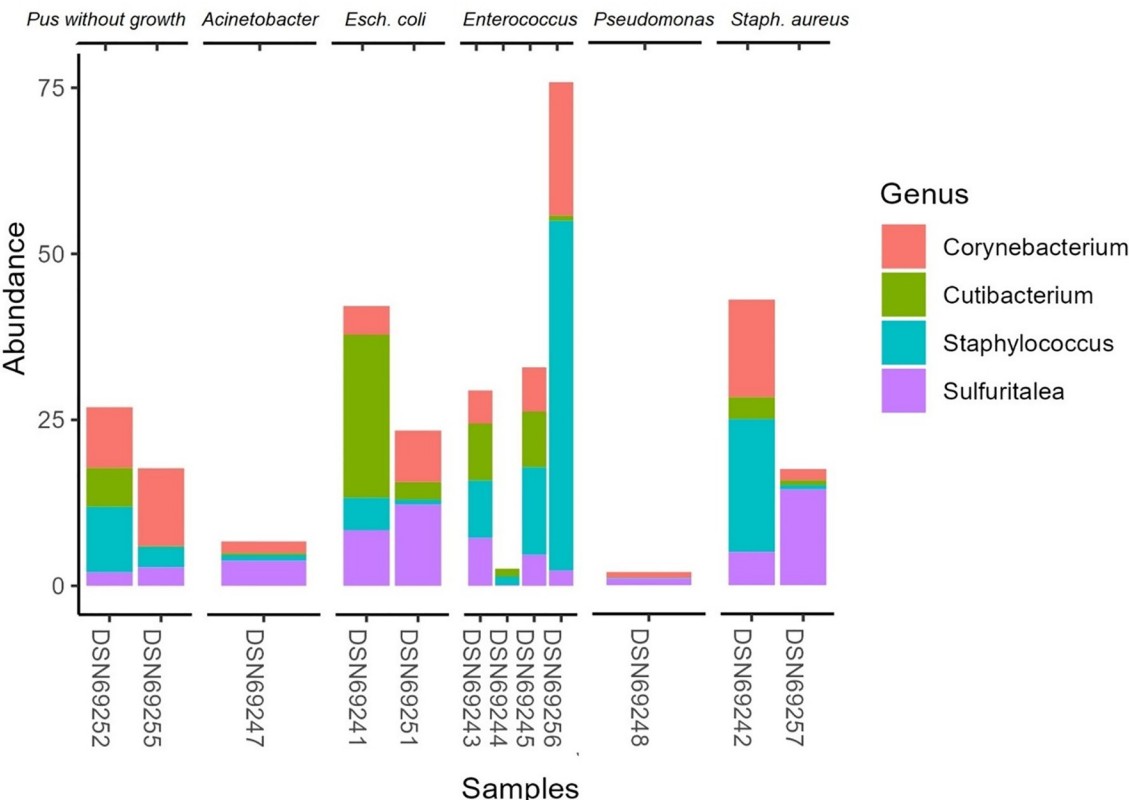

**Fig 18. Relative abundancy of patients' skin microbiota with SSI-isolated microorganisms.**

## Supporting information

**S1 File. Ethical clearance of the DESTINE study at all levels in Uganda.**
(PDF)

**S2 File. Authors' contribution, list of abbreviations, list and description of tables, figures, and supporting information files.**
(PDF)

**S3 File. PLOS One human subjects research checklist.**
(PDF)

**S4 File.**
(DOCX)

## Acknowledgments

The authors acknowledge the participants who kindly accepted to be part of the study. Our special acknowledgment to Mr. Fred Ashaba Katabazi, Mr. Moses Nsubuga Luutu, Mrs. Alice Bayiyaga, Dr. Rose Nabatanzi, Dr. Anthony Fuller, Dr. Tim De Paw, Dr. Sarah Hendrickx, Dr. Tybault Hollanders, Prof. Kalangu Kazadi, Dr. Trésor Kabeya, Sr. Merab Asekene, and the entire Mulago Neurosurgery team, the Neurosurgical Society of Uganda (NSU), and la Société Congolaise de Neurochirurgie (SCNC) for their contributions to this research project in

different form support such as expertise consultation, proof-reading, etc. HML acknowledges the previous support from the *Else-Kröner-Fresenius-Stiftung* through the BEBUC Excellence Scholarship Program.

The abstract of this article was presented at the World Federation of Neurological Societies (WFNS) Congress in December 2023 in Cape Town as an oral presentation, and received the WFNS Atos Alves de Sylva Young Neurosurgeon Award 2023, and also as an oral presentation at the AGM of the Association of Surgeons of Uganda (ASOU) in March 2024.

## Author Contributions

**Conceptualization:** Hervé Monka Lekuya, David Patrick Kateete, Jelle Vandersteene, Jean-Pierre Okito Kalala, Moses Galukande.

**Data curation:** Hervé Monka Lekuya, David Patrick Kateete, Edgar Kigozi, Larrey Kasereka Kamabu, Rose Nantambi, Jean-Pierre Okito Kalala.

**Formal analysis:** Geofrey Olweny.

**Funding acquisition:** Hervé Monka Lekuya, Jean-Pierre Okito Kalala, Moses Galukande.

**Investigation:** Hervé Monka Lekuya, Geofrey Olweny, Edgar Kigozi, Rose Nantambi.

**Methodology:** Hervé Monka Lekuya, Fredrick Makumbi, Jelle Vandersteene, Jean-Pierre Okito Kalala, Moses Galukande.

**Project administration:** Hervé Monka Lekuya, Rose Nantambi, Jean-Pierre Okito Kalala, Moses Galukande.

**Resources:** Hervé Monka Lekuya.

**Software:** Geofrey Olweny.

**Supervision:** David Patrick Kateete, Fredrick Makumbi, Stephen Cose, Jelle Vandersteene, Edward Baert, Jean-Pierre Okito Kalala, Moses Galukande.

**Validation:** Hervé Monka Lekuya, David Patrick Kateete, Geofrey Olweny, Edgar Kigozi, Stephen Cose, Jelle Vandersteene, Edward Baert, Jean-Pierre Okito Kalala, Moses Galukande.

**Visualization:** Hervé Monka Lekuya, Jelle Vandersteene, Jean-Pierre Okito Kalala, Moses Galukande.

**Writing – original draft:** Hervé Monka Lekuya.

**Writing – review & editing:** David Patrick Kateete, Geofrey Olweny, Larrey Kasereka Kamabu, Safari Paterne Mudekereza, Rose Nantambi, Ronald Mbiine, Fredrick Makumbi, Stephen Cose, Jelle Vandersteene, Edward Baert, Jean-Pierre Okito Kalala, Moses Galukande.

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
