## [Decision Letter · Decision Letter 0]

9 Jun 2024

PONE-D-24-15717Metagenomic sequencing of the skin microbiota of the scalp predicting the risk of surgical site infections following surgery of traumatic brain injury in sub-Saharan Africa.PLOS ONE

Dear Dr. Lekuya,

Thank you for submitting your manuscript to PLOS ONE. After careful consideration, we feel that it has merit but does not fully meet PLOS ONE’s publication criteria as it currently stands. Therefore, we invite you to submit a revised version of the manuscript that addresses the points raised during the review process.

We look forward to receiving your revised manuscript.

Kind regards,

Arghya Das, MD

Academic Editor

PLOS ONE

Journal Requirements:

2. Thank you for stating the following financial disclosure: "Makerere Research Innovation Funds (Mak RiF) from the Government of Uganda, and Special Research Funds (BOF funding) of the Ghent University from the Flemish Government under the DESTINE Study."

Additional Editor Comments:

INTRODUCTION

Lines 45-46: One of the post-operative challenges of the surgical management of traumatic brain injury (TBI) is the occurrence of surgical site infections (SSI) in sub-Saharan Africa (SSA),[1-3]

Comment: None of the three cited references actually mentioned the study findings of sub-Saharan Africa.

Please replace it with more suitable references.

Lines 47-49: This post-infectious complication is the commonest morbidity that leads to subsequent postoperative mortality among TBI patients up to 3 months after the initial surgery, with a prolonged hospitalization stay, increased healthcare costs, and impaired patient outcomes [4].

Comment: The cited reference is almost four decades old. Please cite a more suitable and latest reference.

Lines 49-52: The bacterial origin of the infection during surgical procedures is very complex. They can be endogenous, exogenous (contamination), or both. Frequently, the infection originates from the surrounding residual bacteria of the skin of the scalp where the surgical incision is made [5-7].

Comment: The cited reference number 5 mentions specifically mentioned about SSI after cesarean section. The other two studies did not mention anything on SSI. Please cite more suitable references or modify the text without specifying scalp SSI.

Line 64: …….in normal circumstances…..

Comment: The above part of the sentence seems redundant and misleading since biofilm formation may not be considered normal.

Lines 65-67: In case of any change in the micro-environment of the body, a shifting paradigm of the role of local bacteria to common microbial infections up to 65% [6, 10].

Comment: The above sentence is misleading. The cited reference number 10 actually mentioned that biofilms may be responsible for up to 65% of all infections.

Lines 77-78: 16S RNA sequencing

Comment: Please mention ‘16S rRNA sequencing’.

Lines 84-88: The uniqueness of this research…… is more prone or protective against skin infections.

Comment: It appears that the authors tried to justify their study findings, not only the need for the study in the above sentences. Therefore, these lines should better be suited as part of the DISCUSSION and may be moved to the latter part of the manuscript.

MATERIALS AND METHODS

Line 94: DESTINE-study

Comment: Please mention any national or international registry platform where the study may have been registered and mention the registration number.

Lines 95-102: This present study…………………steroid treatment, or with comorbidities.

Comment: Two subheadings, ‘Inclusion’ and ‘Exclusion’ criteria, may be created for this paragraph for better understanding of readers.

Line 99:…hemodynamically stable, and whose informed written consent was obtained…

Comment: Given the serious nature of the patients, please clarify whether written consent was only sought from patients or consent from patients’ next to kin was considered (in case patient is unable to provide consent.

Lines 103-104: They all received………….the team readiness

Comment: This is a crucial part and needs to be elaborated. Please mention the types/names of antibiotics used for prophylaxis. How long was the prophylaxis continued? These information is crucial because they have a direct effect on the clinical outcomes being considered in the present study.

Line 109: abundancy

Comment: Please replace the above word with ‘abundance’. Also make similar changes elsewhere in the manuscript.

Lines 113-114: Collection of the skin swab: During the perioperative period, just after obtaining informed consent, a skin swab of the surrounding normal skin.

Comment: It is imperative to mention the duration of stay in the hospital prior to the collection of skin swabs during the peri-operative period. This is particularly important since a prolonged stay in the hospital may adversely affect scalp colonization by organisms from the hospital environment.

Line 114: e.g: retro-auricular skin at the hairline

Comment: Is there any specific reason for choosing the retro-auricular skin? Please clarify

Line 121-134: DNA extraction………..16s rRNA sequencing

Comment: This entire paragraph may not be required given the length of the manuscript. Instead, the authors may choose to mention that DNA extraction was carried out following the manufacturer’s recommendation.

Line 138: forward and reverse primers

Comment: Please mention sequences of all forward or reverse primers or upload them as supplementary files.

Line 154: Illumina platform

Comment: Please mention the exact name of the instrument with the model number, manufacturer and country of origin.

Lines 156-159: After sequencing………………….microbial communities present in the samples.

Comment: Please mention all software names with their specific use in the bioinformatic analysis.

Lines 160-165: Bioinformatics

Comment: Please mention all software names with their specific use in the bioinformatic analysis. It is also desirable to include the URL link of the applications within parentheses after the names of the software.

RESULTS

Lines 195-196: Indeed, in addition to the study inclusion criteria, we convened with those 57 patients because their samples were amplifiable with a successful quality check for sequencing.

Comment: The above sentence raises some concerns. Does it actually mean that non-ampilfiablility and/or failure in quality checks are also criteria for exclusion? If so, this should be mentioned as an Exclusion criterion under MATERIALS NAD METHODS.

Lines 204-206: The samples of the 12 patients………………. antibiotic susceptibility

Comment: This investigation is very important and has been totally omitted from the MATERIALS AND METHODS. Please mention it under a separate subheading, ‘Laboratory investigation for diagnosis of SSI’ under materials and methods. Mention the type of antibiotic susceptibility testing performed and the clinical breakpoints to denote susceptible, intermediate, and resistant.

Lines 206-209: We found that mono-infection……….12 patients with SSI.

Comment: Please rephrase the above sentence for clarity.

Replace ‘mono-infection’ with ‘mono-microbial infection’. Please make similar changes elsewhere in the manuscript.

Please mention both Genus and species names for the isolates.

Lines 209-210: One case of mortality due to intracranial SSI………………… associated with Escherichia Coli and Klebsiella pneumoniae.

Comment: Please mention ‘intracranial infection’ in place of ‘intracranial SSI’. Please write species names in lowercase letters. All genera and species need to be italicized. Check all organisms’ names in the manuscript.

Lines 218-220: The phyla of………………………. proportion at adult age.

Comment: It is not clear which age group is considered as the pediatric population in the study. As per Figure 4, it seems to be that there were three patients in the pediatric group and two patients in the adult group. Please clarify.

Lines 238-243: Alpha diversity was measured by……………….. of infected and non-infected.

Comment: I think that the information mentioned in this paragraph is better suited under the Statistical Analysis under MATERIALS AND METHODS.

Line 241: Wilcoxon test

Comment: Please clarify which variant of Wilcoxon test was used.

Line 259: Ochnobacterium

Comment: Probable typo.

Lines 263-264: Co-occurrence network graphs of the skin microbiota and their node taxonomic composition between the 2 groups of interest.

Comment: The above sentence is incomplete or grammatically incorrect.

Lines 269-270: LDA effect size (LEfSe) analysis of between two-group differences in skin microbial abundances for personality traits.

Comment: The above sentence is incomplete or grammatically incorrect.

DISCUSSION

The sub-headings under discussion are not required and should be deleted.

Lines 284-285: We found a higher antimicrobial resistance to common antibiotics, and it is well known from microbiologic studies using the biofilm of the bacterial communities [6, 14].

Comment: The above sentence is misleading and needs to be rephrased for a better understanding of the readers. Please ensure that both the references mentioned within brackets have been cited appropriately.

Line 294: ……….. the 3rd and 4th abundant genera…..

Comment: Please mention ‘respectively’ after ‘genera’.

Lines 321-322: Patients who did not develop showed a more balanced and stable microbiome like in the general skin microbiome as described by Egert et al……

Comment: The above part of the sentence is incomplete or grammatically incorrect.

Line 349: multi-resistant

Comment: Replace ‘multi-resistant’ with ‘multidrug-resistant’.

Lines 352-353: TBI Patients who undergo emergency surgery may take several hours to days without adequate skin preparation due to the presence of the hair.

Comment: The above sentence seems incomplete. Please check the sentence and rewrite for clear understanding.

Lines 392-393: List of abbreviations

Comment: ‘OUT’ should be written as ‘OUT’.

Comments on tables

Table 1

There is a major discrepancy in the absolute numbers and percentage calculation for different genders. The column totals of males and females do not match the total number of patients developing SSI and patients without SSI. Accordingly, the Fisher’s exact test p-value needs to be recalculated.

Table 2: In serial number 10, piperacillin-tazobactam has been written as a single word. Authors may also choose to mention cotrimoxazole as ‘trimethoprim-sulfamethoxazole’.

Comments on figures

A large number of figures are included in the manuscripts. Therefore, trivial figures like Figure 3 may be deleted. What is P1, P2, P3, A1, A2 in Figure 4?

Reviewers' comments:

Reviewer's Responses to Questions

**Comments to the Author**

1. Is the manuscript technically sound, and do the data support the conclusions?

Reviewer #1: Yes

Reviewer #2: Yes

2. Has the statistical analysis been performed appropriately and rigorously? 

Reviewer #1: Yes

Reviewer #2: Yes

3. Have the authors made all data underlying the findings in their manuscript fully available?

Reviewer #1: Yes

Reviewer #2: Yes

4. Is the manuscript presented in an intelligible fashion and written in standard English?

Reviewer #1: Yes

Reviewer #2: Yes

5. Review Comments to the Author

Reviewer #1: The study outlined presents an investigation into the relationship between scalp microbiota composition and the risk of Surgical Site Infections (SSIs) following Traumatic Brain Injury (TBI) surgery. While the study provides valuable insights, there are few lacunae where further clarification or improvement is needed:

1. Environmental contamination: The possibility of contamination with environmental flora is not considered, especially when the target population are admitted after potential RTC which may introduce bias into the scalp skin microbiota composition and their representativeness of scalp microbiota diversity. Additionally, the choice of cleaning the sampling sites to get rid of sand or mud, etc. should be addressed.

2.Sample Size and Generalizability: The sample size of the study is not explicitly justified or powered. Given the complexity of microbiota analysis and the potential variability in SSI occurrences, a larger sample size would enhance the study's statistical power and generalizability of the findings. The study acknowledges its limitation of a relatively small sample size from a single center. This raises concerns about the generalizability of the findings to broader populations. Larger multi-center studies would provide more robust evidence.

3. Causal Inference: While the study suggests an association between scalp microbiota dysbiosis and SSI risk, it does not establish causality. Further mechanistic studies are needed to elucidate the underlying biological mechanisms driving this association.

Addressing these lacunae would strengthen the study's design, rigor, and relevance, enhancing the reliability and significance of its findings.

Overall, based on the information provided, the manuscript appears to be technically sound, and the data support the conclusions drawn by the researchers.

Reviewer #2: The manuscript is well written. The authors have throughly researched the impact of scalp microflora on the infections following traumatic brain injury. The manscript can be accepted in the current form.

6. PLOS authors have the option to publish the peer review history of their article (what does this mean?). If published, this will include your full peer review and any attached files.

Reviewer #1: No

Reviewer #2: No

---

## [Author Response · Author response to Decision Letter 0]

20 Jun 2024

Dear Editor,

Dear Reviewers,

Thank you very much for the peer-review feedback. I salute the detailed review and comments on our manuscript, and mostly the final inputs that we have brought have improved the overall quality of the manuscript.

Below, we are addressing the additional Editorial and Reviewers’ comments as requested by the journal guidelines. Please find the point-by-point responses to your queries in the summary table in the attachment.

Best regards,

HL

Corresponding author

---

## [Editor Report · Decision Letter 1]

26 Jun 2024

PONE-D-24-15717R1Metagenomic sequencing of the skin microbiota of the scalp predicting the risk of surgical site infections following surgery of traumatic brain injury in sub-Saharan Africa.PLOS ONE

Dear Dr. Lekuya,

Thank you for submitting your manuscript to PLOS ONE. After careful consideration, we feel that it has merit but does not fully meet PLOS ONE’s publication criteria as it currently stands. Therefore, we invite you to submit a revised version of the manuscript that addresses the points raised during the review process.

We look forward to receiving your revised manuscript.

Kind regards,

Arghya Das, MD

Academic Editor

PLOS ONE

Journal Requirements:

Additional Editor Comments:

Although the authors have satisfactorily addressed the comments and revised the manuscript, few minor technical corrections are still required to make the manuscript nearly flawless.

Please consider the following suggestions while preparing the revised version. Please note the following page and line numbers reflect the same of the clean copy of the revised manuscript.

Page 3, Line 51-52: Frequently, the infection originates from the surrounding residual bacteria of the skin of the scalp where the surgical incision is made [5, 6].

Comment: Authors responded to the comment on original submission that they have modified the text from lines 49-52 by removing the word "scalp" and making the statement applicable to all infections originating from the skin. However, the same is not reflected in the revised manuscript.

Page 3-4, Line 66-67: Indeed, in case of any change in the micro-environment of the body, a shifting paradigm of the role of local bacteria to potential microbial infections [10, 11].

Comment: The sentence is still incomplete and requires rephrasing.

Page 6, Line 117: Vancomycine

Comment: Correct the typo.

Additionally, please write names of all antibiotics within sentence in lower case letters only (including the first letter).

Page 6, Line 120-121: Postoperatively, they also received additional intravenous antibiotherapy in continuation or adjustment in case of evidence of infection........

Comment: The above statement raises concern and require further clarification. What do these words 'continuation or adjustment' refer to? Does it mean that antibiotics were continued even after the surgical intervention (antIbiotic prophylaxis) in all patients, irrespective of the risk or, occurrence of SSI? PLEASE CLARIFY

Also, replace 'antibiotherapy' with 'antibiotic treatment'.

Page 7, Line 135: diagnosis

Comment: Add the word 'clinical' before 'diagnosis'.

Page 8, Line 196: cutadapt

Comment: 'cutadept' to be written as 'Cutadept'.

Page 16, Line 322: An igragh.degree

Comment: Probable typo

Page 20, Line 411-414: TBI Patients who undergo emergency neurosurgical intervention may take several hours to days without adequate incisional site preparation on the skin, and again this is worsened by a complex environment of the densely hairy region of the head.

Comment: The intent of the above sentence is still not clear. Why or for what TBI patients may take several hours? Please specify.

Table 1: There discrepancy in the absolute numbers and percentage calculation for different genders in the table is yet not sorted in the revised manuscript. Please note that as per the statistics in the table, there were 2 Females and 9 Males (totalling to 11 patients) under the SSI (Yes) group. But the actual number of patients in SSI(Yes) group was 12.

Similarly, there were 4 Females and 42 Males (totalling to 46 patients) under the SSI (No) group. But the actual number of patients in SSI(Yes) group was 45.

Please make correction, and do re-calculation of percentages and p-value.

Also, in table 1, the statistical analysis done and mentioned for 'Interval between injury & skin swab collection' does not suit. Fisher's exact test mentioned for the mean(+SD ) days does not seem appropriate. Authors may mention this newly added information only in the text, if it is not exactly fitting in Table 1.

Table 2: Please split or hyphenate 'piperacillintazobactam' which is still mentioned as a single word in the row for patient serial no. 10.

---

## [Author Response · Author response to Decision Letter 1]

27 Jun 2024

Dear Editor,

Dear Reviewers,

Thank you very much for the peer-review feedback. In the attachment, we are addressing the additional Editorial and Reviewers’ comments as requested. Please find the point-by-point responses to your queries in a summary within.

Best regards,

Lekuya

---

## [Editor Report · Decision Letter 2]

4 Jul 2024

Metagenomic sequencing of the skin microbiota of the scalp predicting the risk of surgical site infections following surgery of traumatic brain injury in sub-Saharan Africa.

PONE-D-24-15717R2

Dear Dr. Lekuya,

We’re pleased to inform you that your manuscript has been judged scientifically suitable for publication and will be formally accepted for publication once it meets all outstanding technical requirements.

Kind regards,

Arghya Das, MD

Academic Editor

PLOS ONE

Additional Editor Comments (optional):

'igraph.degree' is still wrongly written as 'igragh.degree' in the revised manuscript.

However, this small correction may be made at the final check stage of the manuscript by authors.
---

## [Editor Report · Acceptance letter]

15 Jul 2024

PONE-D-24-15717R2 

PLOS ONE

Dear Dr. Lekuya, 

I'm pleased to inform you that your manuscript has been deemed suitable for publication in PLOS ONE. Congratulations! Your manuscript is now being handed over to our production team.

Kind regards, 

on behalf of

Dr. Arghya Das 

Academic Editor

PLOS ONE